# NgIohTuned, a new Black-Box Optimization Wizard for Real World Machine Learning

## Abstract

Inspired by observations in neuro-control and various reproducibility issues in machine learning black-box optimization, we analyze the gap between real-world and artificial benchmarks. We (i) compare real-world benchmarks vs artificial ones, emphasizing the success of Differential Evolution (DE) and Particle Swarm Optimization (PSO) in the former case (ii) propose new artificial benchmarks including properties observed in the real world and in particular in neural reinforcement learning, with a special emphasis on the scaling issues, where scale refers to the unknown distance between the optimum and the origin (iii) observe the good performance of quasi-opposite sampling and of Cobyla in some problems for which the scale is critical (iv) observe a robust performance of discrete optimization methods focusing on an optimized decreasing schedule of the mutation scale (v) design more efficient black-box optimization algorithms that combine, sequentially, optimization algorithms with good scaling properties in a first phase, then robust optimization algorithms for the middle phase, followed by fast convergence techniques for the final optimization phase.

All methods are included in a public optimization wizard, namely NgIoh4 (without taking into account the type of variables) and NgIohTuned (taking into account all conclusions of the paper, including taking into account the real-world nature of a problem and/or that it is neurocontrol).

## 1 Introduction

Black-box optimization is the optimization of functions on discrete or continuous domains without any gradient or white-box information. Inspired by the Dagstuhl seminar 23251 *Challenges in Benchmarking Optimization Heuristics* (July 2023), we develop additional benchmarks in a black-box optimization platform, namely Nevergrad, which contains a large family of problems including reinforcement learning, tuning for machine learning, planning and others.

The contributions of this paper are twofold. **First, we analyze benchmarks.** Following Meunier et al. (2022), we observe that one can significantly modify the results of a benchmark by changing the distribution of the optima, in particular by scaling its variables (e.g., by placing them closer to zero or closer to the boundary of the domain). The distribution of the optima, typically induced by the random shifts used in benchmarking platforms, importantly impacts the experimental results. We therefore include a set of different benchmarks with different scaling, ensuring that the tested algorithms cannot be re-parametrized for each specific benchmark. Likewise, we introduce multi-scale and parametric benchmarks for investigating such scaling issues (Section 3.2.2) with the results described in Section 4.1. We also highlight that real-world benchmarks bring new insights, including elements related to budget pointed out by (Ungredda et al., 2022; Dagstuhl participants, 2023) , and showed by the results in the (non-Nevergrad) real-world cases (Appendix G) and in the real-world part of the benchmarking suite of Nevergrad (Section 4.2), with the success of PSO, DE and wizards using them. The diversity of benchmarks is important for making results transferable to other problems (Section 3.1), so we increase the diversity of the distributions of optima over many benchmarks, with a multi-scale approach, and the diversity of the budget/dimension ratio.

**Second, algorithm design.** In Section 3.3, we focus on algorithms which perform well in a context of unknown scale, in discrete, continuous domains, and real-world scenarios. These contributions are integrated into a state-of-the-art wizard for black-box optimization which improves the state of the art on average in many benchmarks (Section 4 and later): NgIoh4, which incorporates our improvements regarding scale issues, and NgIohTuned, which incorporates all the modifications that we propose, including switching to quasi-opposite PSO in neuro-control and to NGOptRW for other real-world problems.

## 2 State of the art

Black-box optimization is an important part of AI, with applications, among many others, in reinforcement learning and planning. The scale, in the sense of the distance to the optimum, has been identified as a key issue in many papers. After Rechenberg (1973) focusing on the adaptation of the scale, Rahnamayan et al. (2007) focuses on initializing population-based methods for robustness to the scale in the continuous context, and in the discrete case, Doerr et al. (2019); Einarsson et al. (2019); Doerr et al. (2017); Dang & Lehre (2016) are entirely based on scheduling the scale of mutations. Methods focusing on a fixed schedule are particularly robust in the discrete setting. In particular, our results confirm their robustness compared to adaptive methods such as the one described in Kruisselbrink et al. (2011a). In terms of continuous black-box optimization methods, Differential Evolution (DE) (Storn & Price, 1997) and Particle Swarm Optimization (PSO) (Kennedy & Eberhart, 1995) are well-known. Compared to CMA (Hansen & Ostermeier, 2003), their focus on quickly approximating the right scale are compatible with very high dimensional settings, where CMA is mainly robust to conditioning/rotation issues. Bayesian methods (Jones et al., 1998) and methods based on machine learning are another branch of the state-of-the-art: among them, SMAC3 (Lindauer et al., 2022) and HyperOpt (Bergstra et al., 2015) perform particularly well. Cobyla (Powell, 1994) comes from the mathematical programming community, and it frequently performs remarkably well in low budget cases (Raponi et al., 2023). Sequential Quadratic Programming is another well known approach with an excellent local convergence rate. Recently, wizards (inspired by other areas such as (Xu et al., 2008)) have become usual. These tools combine various base algorithms, for being immediately (without tuning) reasonably effective on many benchmarks, independently of noise, parallelism, budget, types of variables, and number of objectives. They typically use a lot of static portfolio choices (Liu et al., 2020; Meunier et al., 2022) and of bet-and-run (Weise et al., 2019). We use chaining more intensively than existing wizards. We note that the best performing method in the BBO challenge (AX-team, 2021) is a wizard termed Squirrel (Awad et al., 2020) combining, among others, DE and SMAC3.

In terms of platforms, many libraries exist (e.g., (Johnson, 1994; FacebookResearch, 2020; Virtanen et al., 2020)). Nevergrad (Rapin & Teytaud, 2018) imports these libraries and others. In terms of benchmarks/applications, the BBO Challenge (AX-team, 2021) (close to real-world, with best performance obtained by a wizard including differential evolution, in Awad et al. (2020)), Keras (Chollet et al., 2015), scikit-learn (Pedregosa et al., 2011), COCO/BBOB (Hansen et al., 2009a) (artificial, best performance by CMA variantsHansen & Ostermeier (2003)), LSGO (Li et al., 2013), IOH (Doerr et al., 2018), OpenAI Gym (Brockman et al., 2016) are well known. Nevergrad includes them or some of their variants and many others, and, with our present work, including quasi-opposite forms of DE, SQOPSOZhang et al. (2009), NgIoh wizards, Carola algorithms (see Section 3.3 for these algorithms) and new benchmarks including benchmarks with multiple scales (Section 3.2.2).

## 3 Experimental Setup

Motivated by recent warnings such as (Kapoor & Narayanan, 2023; Li & Talwalkar, 2019), we first take a moment in Section 3.1 to reflect on reproducibility, before we present selected benchmark suites (Section 3.2) and algorithms (Section 3.3). Concerning the suites and algorithms, we cannot provide here an exhaustive presentation of all methods. Hence, we present the most relevant ones and refer interested readers to (Rapin & Teytaud, 2018). Implemented in Python programming language, the Nevergrad platform can be considered human-readable.

### 3.1 Reproducibility

Reproducibility matters (López-Ibáñez et al., 2021). All our code is hence available in open access. It is now part of the Nevergrad codebase (Rapin & Teytaud, 2018). A PDF with all experimental results is available at `tinyurl.com/dagstuhloid`. Though our focus is on the *ability to rerun everything*, the entire data is available at `tinyurl.com/bigdagstuhloid`[1]. As these URLs are automatically updated, they might differ thanks to additional work by contributors and re-runs. Upon acceptance of this submission, we will make a "frozen" version of code and data and store them into a permanent storage facilities such as Zenodo. In the same vein, the version of (e.g., Pypi) packages can have an impact, but maybe results which are valid only for a specific version of some packages might not be that valuable. As most platforms, Nevergrad requires a minimum version number for each package, and not a fixed version number. Our modifications do not change this. Details about reproducibility are mentioned in Appendix B.

### 3.2 Benchmark Suites (a.k.a. Problem Collections)

We seek to have a diverse set of benchmark suites, covering large ranges of problem settings encountered in practice. This includes, for instance, diversity with respect to budget, performance measure, distribution of the optima. Table 1 summarizes the diversity of our benchmarks and their parameters. For each benchmark suite, the detailed setup is described at `tinyurl.com/2p8xcdrb`

#### 3.2.1 Budgets

Ungredda et al. (2022); Dagstuhl participants (2023) showed that cases with budget with more than 100 times the dimension might be the exception rather than the norm. In real-world applications, we may even face settings in which the total number of function evaluations may not exceed a fraction of the dimension. We therefore consider a large variety of different scalings of the budget, including cases with budget far lower than the dimension.

#### 3.2.2 Scaling and Distribution of Optima in Continuous Domains

Throughout the discussion, we assume that the center of the domain is zero. This is not the case in all benchmarks: this is just a simplification for shortening equations, so that we can use $-x$ for symmetries instead of $c - (x - c)$ with $c$ being the center, and $||x||$ instead of $||x - c||$. We observe that scaling is an important issue in benchmarks. Typically, in real-world scenarios, we do not know in advance the norm of the optimum (Meunier et al., 2021; Kumar, 2017). Assuming that the optimum has all coordinates randomly independently drawn with center zero implies that the squared norm of the optimum is, nearly always, close to the sum of variances: this is the case in many artificial benchmarks. Consequently, it reduces the generality of the conclusions: conclusions drawn on such benchmarks are valid essentially on problems for which there is a nearly constant norm of the optimum.

**Different distributions of the optimum: MS-BBOB.** MS-BBOB is quite similar to BBOB (Hansen et al., 2009b) or YABBOB (Rapin & Teytaud, 2018). However, MS-BBOB (multi-scale black-box optimization benchmark), has different scales for the distribution of optima. This is done by introducing a scaling factor $\tau$ which varies in $\{0.01, 0.1, 1.0, 10.0\}$. This scaling factor is used as a factor for the random drawing of optima. For example, in some benchmarks, Nevergrad uses a normal random variable for choosing the optimum. Thus, we multiply this random variable by $\tau$.

**Zero-penalization: ZP-MS-BBOB.** Many benchmarks, including our benchmarks in MS-BBOB are symmetrical w.r.t. zero. The optimum might be translated, but that translation has zero mean. This special role of the center might imply that the neighborhood of zero provides too much information. Actually, many real-world problems have misleading values close to zero, in particular in control or neuro-control (e.g., for neuro-control the control is just zero if all weights in a layer are zero). Therefore, we consider zero-penalized problems, with a heavy penalty for candidates much closer to zero than the optimum. We call this variant ZP-MS-BBOB (zero-penalized MS-BBOB). We note that PSO variants perform quite well in these benchmarks, which coincides with the results described in Raponi et al. (2023).

---

[1]Warning: $> 300$MB, representing data from more than 20 million runs.

**Real-world benchmarks.** "(RW)" means that the benchmark is a real-world problem. Note that the definition of "real-world" is not so simple. We are entirely in silico, and in some cases the model has been simplified. This just means that we consider this as sufficiently real-world for being tagged that way. Our experiments include neuro-control with OpenAI Gym (Brockman et al., 2016), policy optimization with Aquacrop (Raes et al., 2009), PCSE (de Wit, 2021), and hyperparameter tuning with Keras (Chollet et al., 2015) and scikit-learn (Pedregosa et al., 2011). Note that an additional real-world benchmarking is performed in appendix G, for checking the validity of our conclusions on completely distinct problems outside Nevergrad.

### 3.3 Key Algorithms for scaling issues

We highlight here only a few selected algorithms. All details and implementations of the algorithms discussed here and many more are available at `github.com/facebookresearch/nevergrad`.

#### 3.3.1 Opposite and quasi-opposite sampling

Rahnamayan et al. (2007) propose to initialize the population in DE as follows: (i) randomly draw half the population as usual and (ii) for each point $p$ in this half population, also add $-p$ (opposite sampling) or $-r \times p$ (quasi-opposite sampling, where $r$ is chosen i.i.d. uniformly at random in the interval $[0, 1]$). A key advantage of the quasi-opposite method is that the resulting population includes points with all norms, which is beneficial for settings with unknown scaling. We use quasi-opposite sampling in DE and PSO, with variants termed QODE, QNDE, SPQODE, LQODE, SODE, QOTPDE, QOPSO, SQOPSO, fully described in Appendix C. SQOPSODCMA and SQOPSO are followed by diagonal CMA. We observe good results, overall, for SQOPSO and various quasi-opposite tools (Section 5), in particular in the real-world context (Section 4.2).

#### 3.3.2 Other algorithms focusing on scaling in the continuous case

Cobyla is good when the scale of the optimum is unknown, as shown by later results, and quasi-opposite sampling helps DE in the same context. Another solution for guessing the scaling of the optimum is to assume that the scaling of the optimum $x$ for different variables might be similar, i.e., $\log |x_i| \simeq \log |x_j|$ for $i \neq j$. Inspired by this observation, we propose RotatedTwoPointsDE, a variant of DE using a 2-point crossover (Holland, 1975), with the possibility of moving the cut part to other variables. Thus, more precisely, DE typically mixes the $i^{th}$ variable of an individual and the $i^{th}$ variable of another individual and

---

**Algorithm 1** Three variants of Carola. MetaModel refers to the MetaModel implementation in (Rapin & Teytaud, 2018), based on quadratic approximations built on the best points so far.

| Carola1: | Carola2: | Carola3: |
|---|---|---|
| **Require:** Budget $b$ | **Require:** Budget $b$ | **Require:** Budget $b$, number $w$ of workers |
| Apply Cobyla with budget $b/2$. | ***Fast approximation:*** apply Cobyla with budget $b/3$. | Apply $w$ copies of Carola2 in parallel, with budget $b/w$ |
| Apply CMA with Meta-Model with budget $b/2$ and initial point the best point so far. | ***Robust local search:*** Apply CMA with MetaModel with budget $b/3$ and initial point the best point so far. | |
| | ***Fast local search:*** Apply SQP (Sequential Quadratic Programming) with initial point the best point so far and budget $b/3$. | |

---

the child gets the result at the $i^{th}$ position (Alg. 3). This happens for several indices $i$, but the $i^{th}$ variable has no impact on the $j^{th}$ variable if $j \neq i$. TwoPointsDE uses the two-points crossover, which has a similar property: the difference with the classical DE is that the impacted variables are in a segment of consecutive variables. Both DE and TwoPointsDE find scales by working somehow separately on variables. RotatedTwoPointsDE can move this segment of consecutive variables, and therefore it might combine the $i^{th}$ variable of an individual and the $i^{th}$ variable of another individual and the child gets the result at the $j^{th}$

position where $j = i + k$ (modulo the number of variables) for some $k \neq 0$. The assumption behind Rotated­TwoPointsDE is that the scale is not totally different, at least in terms of order of magnitude, for different variables: we can carry variables from a position to another. GeneticDE, then, uses RotatedTwoPointsDE during 200 candidate generations (for finding the correct scale) before switching to TwoPointsDE. We observe an excellent performance of GeneticDE on specific problems, though this requires further investigation as opposed to quasi-opposite sampling which performs very well on most real-world problems, or as opposed to Carola2 and its integration in the NgIoh4 wizard defined in Section 3.3.4, which performs excellently on many benchmarks as discussed later.

### 3.3.3 The scaling of mutations in the context of discrete optimization

In discrete optimization, the good old $1/d$ mutation consists in randomly mutating each variable with probability $1/d$ in dimension $d$. Typically, a single variable is mutated; and it rarely includes more than two variables. Some algorithms, in particular after the good results described in Dang & Lehre (2016), use a fixed random distribution of mutation rates. The adaptation of FastGA (Doerr et al., 2017) in Nevergrad consists in randomly drawing a probability $p$ (instead of using $p = 1/d$) in $[0, \frac{1}{2}]$ (in $[0, 1]$, if the arity is greater than two). DiscreteLenglerOnePlusOne, inspired by Einarsson et al. (2019), consists in using a schedule. In this case, the probability $p$ decreases during the optimization run. We observe good results for DiscreteLenglerOnePlusOne: for example for the Bonnans benchmark (Bonnans et al., 2023), which is completely different from the functions used for testing and designing DiscreteLenglerOnePlusOne that is mathematically derived on simpler functions. The results are presented in Figure 1. Results of

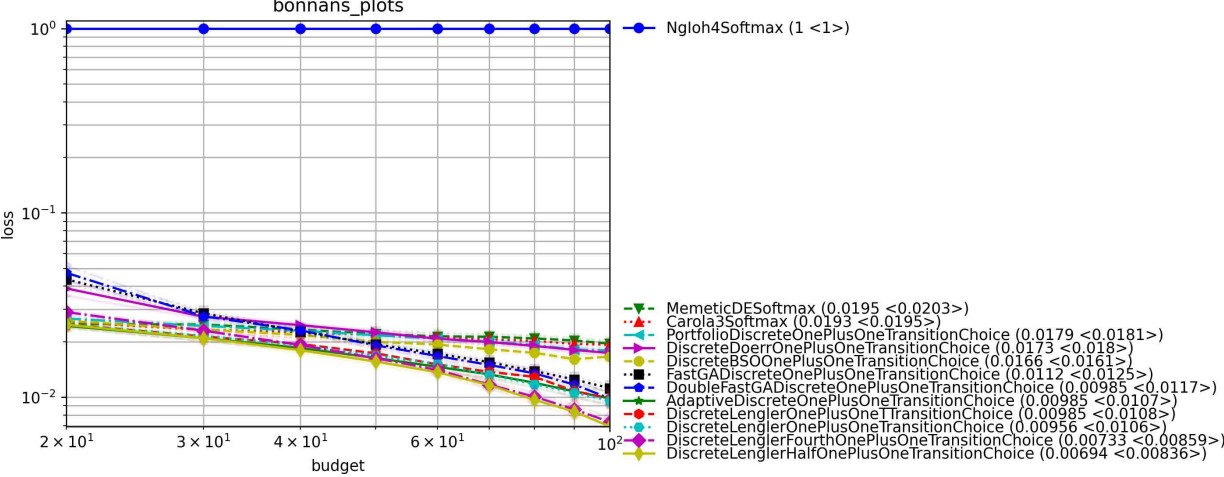

Figure 1: Various methods (86 algorithms run; only the best 12 ones are presented, and the single worst) on the Bonnans (discrete) function. The Softmax representation (converting the problem to a continuous one as optionally proposed in (Rapin & Teytaud, 2018)) performs poorly here compared to the standard TransitionChoice. The DiscreteLenglerOnePlusOne method (and its variant with modified parameters, with similar names) performs well on Bonnans functions (Bonnans et al., 2023).

DiscreteLenglerOnePlusOne are also good on InstrumDiscrete, SequentialInstrumDiscrete, and PBOReduced problems. In terms of ablation, most of the variants with perturbed hyperparameters also perform well.

### 3.3.4 Chaining for the scale in the continuous case: Carola algorithms, and the NgIoh wizard including them.

From observations on IOH (Doerr et al., 2018), we propose two new principles for the design of black-box optimization wizards. NGOpt is the current wizard of Nevergrad, and NGOptRW is another wizard doing a bet-and-run between DE, PSO and NGOpt during 33% of the budget before switching to the best of them (unless the benchmark is noisy or discrete, in which case it simply uses NGOpt). First, whereas it is classical (e.g., Memetic algorithms (Moscato, 1989)) to run evolution strategies first and local methods

afterwards (as Nevergrad's NGOpt frequently does), we observe that Cobyla is excellent for low budget. We therefore consider Carola (Cost-effective Asymptotic Randomized Optimization with Limited Access), a method running Cobyla first and then other methods, as presented in Algorithm 1. Second, our insights are gathered in a new black-box optimization wizard, which we dub NgIoh. We demonstrate in this work that this wizard performs well, on average, on many benchmarks. Overall, including the many benchmarks on which NgIoh does not differ too much from NGOpt, NgIoh slightly outperforms the existing wizard NGOpt from Nevergrad, with a clear gap specifically for problems such that the scale of the optimum cannot be known in advance (Section 4.1). We build several variants, which perform similarly. NgIoh4 performs slightly better overall. NgIoh4 is basically the same as NGOpt (Nevergrad's wizard), except that it switches to Carola2 depending on rules that prefer the Carola2 algorithm in case of moderate budget (Algorithm 2). The constants in the rules were chosen based on the observations described in Doerr et al. (2018). The different variants (performing similarly) are available in Rapin & Teytaud (2018). NgIoh4 is slightly better.

---

**Algorithm 2** The NgIoh4 pseudocode, combining NGOpt and ideas extracted from results in IOH(Doerr et al., 2018).

---

**Require:** Budget $b$, dimension $d$, domain $D$, number $w$ of workers.
  **if** $w = 1$ and $D$ is continuous and ($d < 100$ and $20d \leq b \leq 1000d$) or ($d < 50$ and $b < 1000d$). **then**
    Apply Carola2
  **else**
    Apply NGOpt
  **end if**

---

**Ablation:** Carola3 is an adaptation of Carola2 for the parallel case, so let us focus on the comparison between Carola1, Carola2, and algorithms on which they are based, namely Cobyla, CMA and MetaModel. We observe in Figure 2 better results for Carola2. MetaModel and several CMA variants are absent of the figure because we keep only the 25 best of the 57 tested methods: CMA, OldCMA (before some tuning), LargeCMA (with larger initialization scale) and MetaModel (CMA plus a quadratic surrogate model, as used as a component of Carola2) are ranked 43, 29, 40 and 33 respectively (vs 3 for Carola2). We also tested several variants of Carola (available in the codebase and visible in some of our plots), without much difference.

### 3.4 Towards real-world wizards: taking into account high-level information

We also define the NgIohTuned wizard (full details in Section H.3). NgIohTuned is similar to NgIoh4, but with tuned parameters and using high-level information such as "real-world problem" (for switching to NGOptRW) and "neural control" (for switching to SQOPSO). Due to this, NgIohTuned uses more information than other algorithms and can therefore not be compared to others in a fair manner, but we include the results in appendix and in Figure 4 (right): NgIohTuned is basically the aggregation of all our conclusions in a single wizard.

### 3.5 Performance Criteria

For each benchmark, we consider two figures. First, the **Frequency of winning figure.** A heatmap, showing the frequency $f_{m,m'}$ at which a method $m$ (row) outperforms on average another method $m'$(column). Frequencies are computed over all instances and all budgets. Methods are then ordered by the average $score_m$ of these frequencies $f_{m,m'}$ over all other methods $m'$. The columns show the names of the methods, appended with the number of settings they were able to tackle (for example, some methods have no parallel version and therefore do not run on all settings).

Second, the **Normalized simple regret figure.** A convergence curve, with the budget on the x-axis and the average (over all budgets) normalized (linearly, to $[0, 1]$) loss. Note that some benchmarks do not have the same functions for the different values of the budget. Therefore, we might have a rugged curve, not monotonous. This is even more the case for wizards such as NGOpt or NGOptRW, which make decisions based on the budget. They might make a bad choice for some values of the budget, leading to irregular curves.

The complete archive (see Appendix B) shows many competence maps. Given the hyperparameters of a benchmark (e.g., dimension, budget, level of noise, among others), the competence maps in the full archive show, for a given pair of hyperparameter values, which algorithms performs the best on average.

## 4 Selected benchmarks

We cannot include here all benchmarks from the Nevergrad platform, including our additions. We present only some of them. First, multiple scale benchmarks because we believe that Section 3.2.2 points out an important issue for improving black-box optimization benchmarks. Second, real-world benchmarks, because we need more benchmarks rooted in real-world problems. We refer to the automatically generated `tinyurl.com/dagstuhloid` for further details and Section 5 for an aggregated view of all the results. In Appendix G, we present external applications, so that more independent results (outside Nevergrad) are included. The appendix contains additional figures for a more extensive view of our results.

### 4.1 Multi-scale black-box optimization benchmarks: dealing with the scaling issues

In the case of continuous optimization, we present new benchmarks, adapted from YABBOB (Rapin & Teytaud, 2018) using comments from Section 3.2.2. While CMA variants dominate in BBOB Hansen et al. (2009a) (small scale, large budget, focus on frequency of solving with a given precision) and DE variants dominate in LSGO (Li et al., 2013) (larger scale, groups of variables), we propose a benchmark close to BBOB or YABBOB, but with a specific effort to not make the scale of the norm of the optimum to be known in advance (Section 3.2.2).

An important hyperparameter for the optimization methods, in particular when the budget is moderate, is the scale. Sometimes the optimum is close to the center, sometimes it is far. The principle of our proposed and open-sourced multi-scale BBOB (MS-BBOB) benchmark is that it contains four different scales, and the algorithms are not informed of which scale is used for each instance. We also apply the zero-penalization, as discussed in Section 3.2.2. The resulting benchmark is termed ZP-MS-BBOB, and experiments are presented in Figure 2: they show the good performance of Carola/NgIoh. Fig. 3 also shows that we do not deteriorate the performance too much on YABBOB (which does not have a multi-scaling) and Fig. 4 shows that we improve results on the complete family of benchmarks (in particular with NgIohTuned, right of Fig. 4). It is an artificial benchmark, but it is inspired by various existing benchmarks (Cotton, 2020a;b; Raponi et al., 2023). The best methods are all based on Carola (Section 3.3.4) or NgIoh (which introduces Carola inside NGOpt), or on quasi-opposite sampling. We conclude that the Carola method (using Cobyla first) and quasi-opposite samplings are both good for adapting a method to the right scaling of variables. Figures 6 and 7 present the results on the variants of BBOB in Nevergrad, showing that results are not deteriorated, compared to NGOpt, on these other benchmarks. Section 5 also shows that, on the wide complete family of benchmarks in Nevergrad, NgIoh4 outperforms NGOpt.

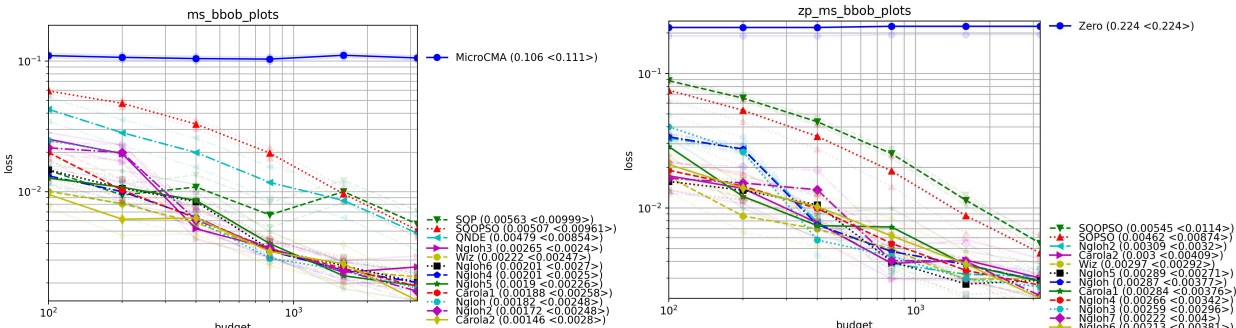

Figure 2: The best performing methods MS-BBOB (left) and for ZP-MS-BBOB (right) for normalized regret. Both: we include the 12 best methods and the worst. NGOpt and CMA are not in the 12 best, e.g. on the right NGOpt is ranked 15th and CMA is ranked 49th (more details in Fig. 17, and more complete results including NgIohTuned in Fig. 16), and an ablation in Fig. 5. The good performances of Carola and NgIoh variants (including Wiz, also based on Carola2) are visible in both. The quasi-opposite variant of PSO is also good. Carola2 is slightly the best of the Carola variants for most budgets (see detailed curves above, or the ranking for the average frequency of winning in Fig. 17).

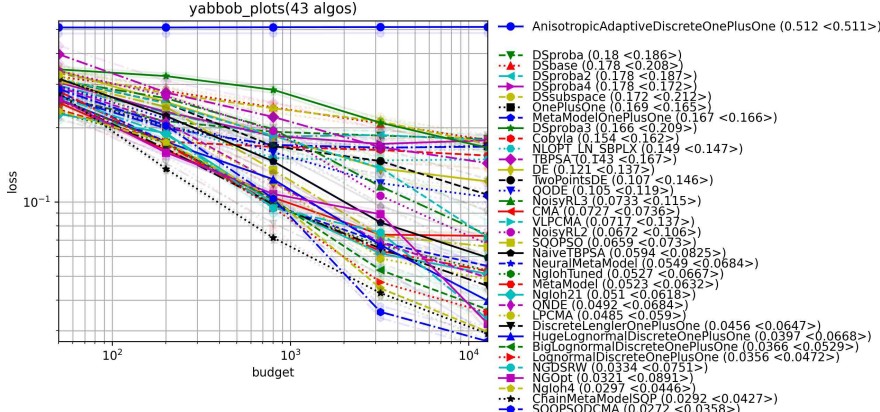

Figure 3: YABBOB, original benchmark: our codes are less impressive on this initial benchmark not equipped with multi-scaling, but they still perform well (though NGOpt is slightly better in particular for lower budget) and in Fig. 4 we observe that on average on many benchmarks NgIoh4 outperforms NGOpt (just thanks to a better scaling) and NgIohTuned outperforms all other methods (by incorporating NgIoh4 and all our conclusions, including the real-world nature and/or neuro-control nature of problems).

## 4.2 Real world benchmarking in Nevergrad

We present in Fig. 4 (left) the number of times each algorithm was ranked best among the list of real-world benchmarks in Nevergrad. Other real world tasks (external to the Nevergrad benchmarks) are available in Appendix G and confirm our conclusions on separated benchmarks.

We note, in the real-world benchmarks of Nevergrad, that PSO and DE variants, in particular with quasi-opposite sampling, perform better than in artificial benchmarks. The rankings below are obtained for a different number of problems for which each algorithm is best in terms of normalized simple regret. We also note that NGOptRW, designed by adapting NGOpt for real-world instances by bet-and-run (Weise et al., 2019) PSO/DE/NGOpt, performs very well. NGOptRW runs these three algorithms for one third of the budget, and keep the best of them for the rest. It vastly outperforms NGOpt and all others, though Carola3 (the possibly parallel adaptation of Carola2) is not bad.

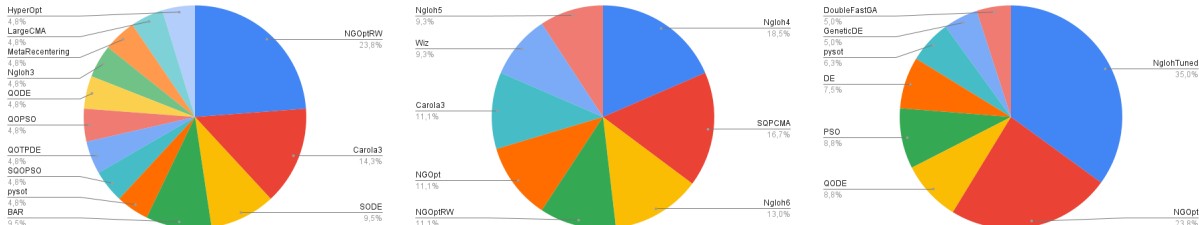

Figure 4: Statistics aggregated over many benchmarks. **Left:** number of times each algorithm is ranked best, on the real-world part of the Nevergrad benchmarks (before adding NgIohTuned): NGOptRW clearly dominates and NGOpt is not very good. **Middle:** number of times each algorithm is ranked best (limited to algorithms ranked 1st at least 5 times), in the complete Nevergrad benchmarks: NgIoh4 performs best (NgIohTuned is still excluded) and outperforms NGOpt, which is better than NGOptRW. **Right:** number of time each algorithm is ranked best in the complete Nevergrad benchmarks, if we remove all our proposed algorithms except NgIohTuned, i.e. we allow our method to use the knowledge "is real-world" or "is a neurocontrol problem": we see that NgIohTuned (combining, by design, the success of NgIoh4 in artificial benchmarks and of NGOptRW in the real-world case or SQOPSO for neurocontrol) outperforms NGOpt and all previous algorithms. Details: SODE, SQOPSO, QODE, QOPSO are defined in Section 3.3.1 and the Appendix C. LargeCMA is CMA with greater initial variance. BAR is a bet-and-run of the $(1+1)$ evolution strategy and DiagonalCMA and OpoDE, where OpoDE runs the $(1+1)$ strategy with one-fifth rule during half the budget followed by differential evolution.

## 5 Statistics over all benchmarks

For shorts, we include below only the number of times each method was ranked first. NgIoh4 performs best with 10 times the first position. Wiz is a variant of NgIoh so that we see a strong domination of NgIoh variants. It is also the best for the number of times it is ranked in the top 2 and for the number of times it is ranked in the top 3. Detailed results are presented in Figure 4.

The full details are reported in Appendix E. NgIoh4 performs even better if we remove the variants Wiz, NgIoh6 and NgIoh5 (documented in Rapin & Teytaud (2018)) from the statistics because they are quite similar.

## 6 Conclusions

**Scale matters.** We note that in both continuous and discrete benchmarks, the scale is important, even more specifically in the black-box case. For the continuous case, besides previously discussed results in Figure 2 and global statistics (Section 5), Appendix H.1 presents detailed comparative results showing how much our tools (Cobyla as a first step and quasi-opposite sampling mainly, and to a lower extent GeneticDE) dedicated to scale work better than previous tools when the scale of the optimum is unknown. **In spite of (actually, even because of) the random shift method, many benchmarks have roughly the same norm of the optimum for all instances.** If we define the position of optima by e.g., a multivariate normal distribution with mean zero and identity covariance matrix (or more generally, independent coordinates with all roughly the same variance, so that variants of the central limit theorem can be applied), then in large dimension the optimum has, almost always, a norm scaling as $\sqrt{dimension}$ (see Section 3.2.2). This is not observed in real-world benchmarks, hence the great real-world performance of the methods above (quasi-opposite sampling) tackling such issues. We advocate MS-BBOB or ZP-MS-BBOB for designing continuous artificial benchmarks close to scaling issues found in the real-world: their results are closer to real-world results (Section 4.2) than other artificial continuous benchmarks in the sense that quasi-opposite sampling or a warm-up by Cobyla are helpful in both cases. More precisely, compare Fig. 4, left and Fig. 2: in both cases methods based on quasi-opposite sampling and warmup by Cobyla (such as Carol* and NgIoh*) perform well, though we also note, independently (see Reality Gap below), good results for DE and PSO and their combination for NGOptRW in real-world settings.

**In the discrete case**, the best methods are frequently based on Lengler (Einarsson et al., 2019), which is based on a predefined schedule of mutation scales. This schedule differs from the classical $1/d$ mutation, in particular in early stages. We note that the mathematically derived Lengler method outperforms some handcrafted methods in Nevergrad based on the same principle of a decreasing rate, and many methods with adaptive mutation rates. It also outperforms mathematically derived methods such as (Doerr et al., 2017; Dang & Lehre, 2016), which use a fixed probability distribution of the mutation rates. We see the chaining of methods with different regimes in continuous domains as analogous to the predetermined schedule of Einarsson et al. (2019) in the discrete case. In any case, both approaches perform well.

**Quasi-opposite sampling.** An unexpected result is the good performance of quasi-opposite sampling (Rahnamayan et al., 2007) (see QODE, QNDE, QOPSO, SQOPSO in Section 5). We adapted it from DE to PSO, with SQOPSO using, for each particle $p$ with speed $v$, another particle with position $-r \times p$ and speed $-r \times v$ (see Section 3.3.1). Equipped with quasi-opposite sampling, DE and PSO perform quite well in the real-world part of our benchmarking suite (Section 4.2 and Appendix G), with particularly good results of SQOPSO in the case of neurocontrollers for OpenAI Gym (confirmed in Appendix 19). A posteriori, this is consistent with the importance of scale.

**Optimization wizards.** As in SAT competitions and as discussed in the Dagstuhl seminar (Hoos, 2023), we observe excellent results for wizards. All methods performing well on a wide range of benchmarks (without tuning for each benchmark separately) are wizards. NgIoh4 is based on NGOpt, a complex handcrafted wizard based on experimental data, and adds insights from the present paper. It performs well on many benchmarks (Section 5). NgIoh4 aggregates many base algorithms. We see in the detailed logs that it uses CMA, DE, PSO, Holland crossover, bandit methods for handling noise, discrete (1+1) methods with mutation rates schedules, meta-models, Cobyla, multi-objective adaptations of DE, the simple (1+1) evolution strategy with one-fifth rule (Rechenberg, 1973) in some high-dimensional contexts, bet-and-run, and others. Our guess is that it could still be improved by ideas from NGOptRW or quasi-opposite sampling, or by tuning its rules in favor of Carola2 or Carola3 in more general cases. Appendix H.3 confirms that our NgIoh4 and other wizards, besides outperforming NGOpt and non-wizard methods on many Nevergrad benchmarks, also outperform it on BBOB/COCO. NgIohTuned, using high-level information on the type of problem and the types of variables, outperforms other wizards and aggregates in a single code all the conclusions in the present section.

**Low-budget optimization, and first part of a chaining in continuous domains.** SMAC3 got better results than other Bayesian Optimization methods. Bayesian Optimization methods are limited to low budget / dimension contexts, and a strong competitor for continuous optimization with low budget is Cobyla (Dufossé & Atamna, 2022; Raponi et al., 2023). We propose to use Cobyla as a warm-up before other methods, because it is good at understanding the global shape of a problem (Sections 3.3.4 and 4.1). Carola2 is a chaining of 3 stages: Cobyla for a fast first approximation, CMA with MetaModel for a robust optimization, and SQP for a final fast local search. It performs very well as a component of NgIoh4, and its counterpart Carola3 (compatible with parallel settings) performs well in many real-world benchmarks (Section 4.2). Chaining was already present in Rapin & Teytaud (2018), with the classical fast local convergence at the end in many cases, and also for noisy optimization, with a classical algorithm (not taking care of noise) as a first step before switching to a real noisy optimization method in the wizard of Meunier et al. (2022): our application for a first step for a mathematical programming algorithm as a first step is new. Appendix H.3 shows that it is also valid on the BBOB/COCO benchmarks.

**Reality gap.** The gap between real world and artificial benchmarks is still large, as shown by the different best algorithms in real-world vs artificial contexts. In particular, in the continuous context, NGOpt/NgIoh dominates the artificial benchmarks whereas a bet-and-run (termed NGOptRW) of DE, PSO, and NGOpt is better in the real-world. Also, quasi-opposite sampling appears to be great for the real-world context, more than for artificial benchmarks based on random shifts. Random shifts with all components of the shift being independent (or other methods than random shifts provided that many coordinates are independent and have roughly the same variance), lead to nearly the same norm of the optimum for all replicas. Our zero-penalized and multi-scale variants of black-box optimization benchmarks (In Section 4.1, due to the random factor applied to all coordinates, the central limit theorem does not apply) are a step in this direction and we plan to add more of such benchmarks. Another element in terms of reality gap is that in the present paper (and

in most works on wizards), we did not use in NgIoh4 information such that "this problem is real world" or "these variables are weights of a neural net", but NgIohTuned does it and performs quite well (Fig. 4, right, and Appendix H). We note that in (AX-team, 2021; Awad et al., 2020), the best performing method was using the names of variables for choosing between different options. Appendix H.2 presents results on real-world benchmarks with more details, confirming the gap between the global statistics (Section 5, dominated by NgIoh4) and the real world case (Sect. 4.2 and G, dominated by NgOptRW). The algorithm NgIohTuned (the only one among our methods which uses high level information about the problem provided by the user, such as "neurocontrol" or "real-world"), modified for switching to NGOptRW in the real world case and to SQOPSO in the neurocontrol case, performs best overall: we include it in Appendix H. Another important point in terms of bridging the reality gap is that including cases with budget far lower than the dimension is also essential (Ungredda et al., 2022).

**Good benchmarks exist, with a lot of diversity (including real-world and artificial, budget $>>$ dimension and budget $<<$ dimension, with and without noise, with applications from completely different fields, see Table 1): they should be used, in particular in machine learning papers.** Reproducibility is a growing concern in machine learning (Kapoor & Narayanan, 2022), specifically in black-box optimization (Markov, 2023; Meunier et al., 2021). In spite of efforts in the 2000s for creating better benchmarks, benchmarks with optimum at zero, or ad-hoc experiments with a heavily tuned method *with parameters optimized for each benchmark separately*, or with completely different initialization distribution for the baselines, are still published in many conferences. A related and slightly more subtle effect is that the scaling of the initialization can easily make baselines pointless and create non-reproducible results.

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

## A    Ablation regarding ZP and MS: the importance of scaling in continuous domains

Fig. 5 presents an ablation of results of Fig. 2.

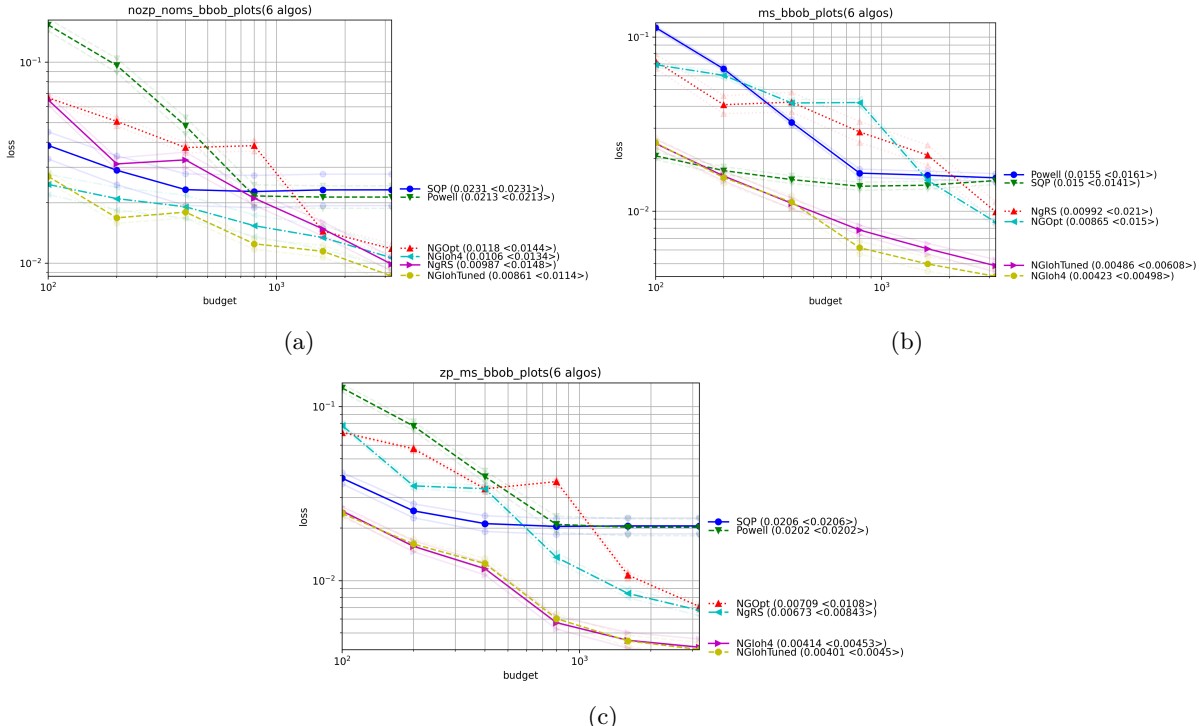

(a)                  (b)

(c)

Figure 5: Ablation for Fig. 2 with the same functions and budgets in all 3 cases, and with a restricted set of algorithms: (a), no ZP and no MS; second (b), we add MS; third (c), we add ZP. While ZP has less impact, we observe that switching from vanilla (a) to MS (multi-scale, b) makes NgIohTuned or NgIoh4 vastly better than NGOpt (still the case after adding ZP, bottom). We believe that such multiple scaling should be part of artificial benchmarks for bridging a part of the gap with real-world benchmarks.

## B    Reproducibility

How to reproduce the results in the present paper:

- Install Nevergrad by cloning the git repository (see details at (Rapin & Teytaud, 2018)).

- Running:
  - Without cluster: `python -m nevergrad.benchmark yabbob --num\_workers=67` if you want to run YABBOB on 67 cores.
  - With cluster equipped with Slurm: Run "sbatch scripts/dagstuhloid.sh" script for launching experiments with Slurm. It is written assuming that Slurm is installed: it should be feasible to adapt it to other job scheduling tools. Running this script several times will increase the number of replicas and increase precision.

- For plotting results, run "scripts/dagstuhloid_plot.sh". Of course, some data might be missing if not enough runs are complete.

- To modify the parallelism, dimension, budget, list of tested algorithms, you might edit `nevergrad_repository/nevergrad/benchmark/experiments.py`.

The present paper in LaTeX is automatically generated by the commands above. Then, the authors have edited the corresponding file for the text and rearranged sections, in particular, moving to the appendix or to an URL many of the individual results on specific benchmarks. An example of the huge original PDF file can be found at `tinyurl.com/dagstuhloid`. We emphasize that reproducibility is not limited to the possibility of reproducing the exact same numbers. We consider results that can only be obtained by certain random seeds uninteresting. We therefore do not fix the seeds.

## C  Additional information on algorithms

We use quasi-opposite DE, in several flavors:

---

**Algorithm 3** The QODE algorithm, with Curr-to-best, $F1 = F2 = .8$, $CR = .5$.

---

**Require:** Budget $b$, population size $p$ ($p = 30$ by default), dimension $d$, objective function $l$ to be minimized

Randomly draw $p/2$ points (uniformly at random by default) $x_1, \ldots, x_{p/2}$.

Define $x_{p/2+i} = -r_i x_i$, with $r_i$ randomly independently uniformly drawn in $[0, 1]$.

Run differential evolution as usual:

**while** Budget not elapsed **do**

    **for** each point $x$ in the population, if the budget $b$ is not elapsed **do**

        Randomly draw $a$ and $b$ in the population, different from $x$

        Let $t$ be the best point so far

        Define $x' = x + F1 * (b - a) + F2 * (t - x)$

        Define, for $1 \leq i \leq d$, $x''_i = x'_i$ with probability $CR$ and $x''_i = x_i$ otherwise.

        Enforce $x''_i = x'_i$ for some randomly drawn $i \in \{1, \ldots, d\}$

        If $l(x'') \leq l(x)$, then replace $x$ by $x''$.

    **end for**

**end while**

---

**Algorithm 4** SQOPSO. By default, $p = 40$, $\omega = 0.5/\log(2)$, $\phi_p = 0.5 + \log(2)$, $\phi_g = 0.5 + \log(2)$.

---

**Require:** Budget $b$, population size $p$, dimension $d$, objective function $l$ to be minimized

Randomly draw $p/2$ points (uniformly at random by default) $x_1, \ldots, x_{p/2}$, and their speeds $v_1, \ldots, v_{p/2}$..

Define $x_{p/2+i} = -r_i x_i$ and $v_{p/2+i} = -r_i v_i$, with $r_i$ randomly independently uniformly drawn in $[0, 1]$.

Initialize $b_i = x_i$ for all $i$ ($b_i$ is the best past position of the $i^{th}$ particle and $g$ the best of the $b_i$

Run particle swarm optimization as usual:

**while** Budget not elapsed **do**

    **for** each point $x_i$ with speed $v_i$ in the population, if the budget $b$ is not elapsed **do**

        **for** each coordinate $1 \leq j \leq d$ **do**

            Randomly draw $r_p$ and $r_g$ in $[0, 1]$

            Update $(v_i)_j = p \times \omega(v_i)_j + \phi_p r_p(b_{ij} - x_{ij}) + \phi_g r_g(g_{ij} - x_{ij})$

        **end for**

        Update $x_i = x_i + v_i$

        If $l(x_i) < l(p_i)$, then update $p_i = x_i$

        If $l(x_i) < l(g)$, then update $g = x_i$

    **end for**

**end while**

---

- QODE, the classical quasi-opposite DE, presented in  Algorithm 3.

- QNDE, which is QODE during half the budget and then BFGS with finite differences.

- SPQODE (SPecial QODE), which is QODE with population size $1 + \sqrt{\log(d + 3)}$ in dimension $d$.

- LQODE (Large QODE), which is QODE with initialization range multiplied by 10 (each individual is multiplied by 10).

- SODE (Special Opposite DE), in which $r$ is $\exp(-5 \times U(0,1))$ instead of $U(0,1)$ (with $U(0,1)$ uniform in $[0,1]$).

- QOTPDE combines TwoPointsDE (DE with Holland 2-points crossover) and QODE.

We also consider quasi-opposite sampling for PSO:

- Randomly draw half the population as usual.

- QOPSO (Quasi-Opposite PSO): for each point $p$ with velocity $v$ in this half population, also add $-r \times p$ with a randomly drawn velocity, with $r$ randomly drawn uniformly in $[0,1]$.

- SQOPSO (Special Quasi-Opposite PSO, defined in Algorithm 4): for each point $p$ with velocity $v$ in this half population, also add $-r \times p$ with velocity $-r \times v$, with $r$ randomly drawn uniformly in $[0,1]$.

## D    Additional information on benchmarks

Table 1: Diversity of our benchmarking platform in Nevergrad and of our automatic report.

(a)

|  | Min | Max |
|---|---|---|
| Dimension | 1 | $20 \times 10^3$ |
| Budget | 10 | $3 \times 10^6$ |
| # objectives | 1 | 6 |
| Noise dissymetries | False | True |
| Noise | False | True$^\star$ |
| # blocks of variables$^\sharp$ | 1 | 16 |
| # of workers | 1 | 500 |

$^\star$many different levels of noise
$^\sharp$with independent rotations

(b)

| Category | Benchmarks |
|---|---|
| Real-world, ML tuning | Keras, Scikit-learn (SVM, Decision Trees, Neural nets) |
| Real-world, not ML tuning | Gym, rockets, energy, fishing, photonics, games |
| Discrete | PBO, Bonnans, others (includes: unordered variables) |
| Continuous, artificial | LSGO, Deceptive, Ya*BBOB |
| Multiobjective | Several problems with 2 to 7 objectives and dim from 2 to 200. |

## E    Statistics over all benchmarks: full details

We point out that NGOpt and its variants are wizards (automatic algorithm selectors and combinators) created by the same authors as Nevergrad, and their (good) results might therefore be biased: we recognize that common authorship for benchmarks and algorithms implies a bias, and, given that our tools are based on NGOpt and other tools in Nevergrad, this applies to us as well. Another issue is that statistics based on frequencies of performing in the top $k$ are a risky thing: when two codes are very close to each other, they are both penalized by each other: we must be careful with interpretations. Nonetheless, we provide aggregated results for convenience.

### E.1    NGOpt versus Base algorithms: validating wizards

Here base algorithms have no metamodel and no complex combinations: wizards are excluded, except NGOpt. NGOpt is the only sophisticated combination: this is an analysis of NGOpt, and this validates that NGOpt performs better than the base algorithms it is built on. We consider statistics on the top $k$ methods, for $k = 1$, $k = 2$, $k = 3$.

### E.1.1    Number of times each algorithm was ranked first: NGOpt and base algorithms

- 29 NGOpt
- 8 HyperOpt
- 8 Cobyla
- 7 QODE

- 6 OnePlusOne
- 5 SODE
- 4 SPQODE
- 4 QOTPDE

### E.1.2   Number of times each algorithm was ranked among the 2 first: NGOpt and base algorithms

- 45 NGOpt
- 16 QODE
- 16 OnePlusOne
- 16 Cobyla

- 12 HyperOpt
- 10 QORealSpacePSO
- 9 SQOPSO
- 8 QOPSO

### E.1.3   Number of times each algorithm was ranked among the 3 first: NGOpt and base algorithms

- 51 NGOpt
- 23 Cobyla
- 20 QODE
- 20 OnePlusOne

- 19 SQOPSO
- 15 HyperOpt
- 14 QORealSpacePSO
- 12 SODE

### E.2   Comparing simple algorithms only: wizards, multilevels, specific standard deviations, and combinations excluded

Simple algorithms might be less overfitted, more robust: we consider the same experiments, but with only "simple" algorithms: no chaining, no metamodel, no tuned parameters, no bet-and-run, no wizard. The success (robustness) of quasi-opposite sampling (for PSO or DE) is visible. in results below. We also note the excellent performance of Cobyla, thanks to great results for moderate budget.

### E.2.1   Number of times each algorithm was ranked first: no wizard, no combination

- 14 Cobyla
- 9 QODE
- 8 OnePlusOne
- 8 HyperOpt

- 6 QORealSpacePSO
- 5 SODE
- 5 QNDE
- 4 SPQODE

### E.2.2   Number of times each algorithm was ranked among the 2 first: no wizard, no combination

- 17 QODE
- 17 OnePlusOne
- 17 Cobyla
- 13 QORealSpacePSO

- 12 SQOPSO
- 12 HyperOpt
- 10 GeneticDE
- 10 DiscreteLenglerOnePlusOneT

### E.2.3   Number of times each algorithm was ranked among the 3 first: no wizard, no combination

- 23 QODE
- 23 Cobyla
- 20 SQOPSO
- 20 OnePlusOne

- 16 QORealSpacePSO
- 15 HyperOpt
- 15 DiagonalCMA
- 13 OldCMA

### E.3   Everything included

For the results of this section, we include all codes, wizards as well as base algorithms. All strong methods are wizards, except tools based on quasi-opposite samplings. The only algorithms making it to the top are (i) wizards (ii) bet and run / aggregations (such as SQPCMA) (iii) HyperOpt (iv) quasi-opposite tools (v) Carola variants.

#### E.3.1   Number of times each algorithm was ranked first: everything included

- 10 NgIoh4
- 9 SQPCMA
- 7 NgIoh6
- 6 NGOptRW

- 6 NGOpt
- 6 Carola3
- 5 Wiz
- 5 NgIoh5

#### E.3.2   Number of times each algorithm was ranked among the two first: everything included

- 22 NgIoh4
- 14 NgIoh5
- 12 NgIoh6
- 11 SQPCMA

- 11 NGOpt
- 10 Shiwa (an old wizard, anterior to NGOpt, designed in Liu et al. (2020))
- 8 QODE
- 8 NgIoh2

#### E.3.3   Number of times each algorithm was ranked among the three first: everything included

- 29 NgIoh4
- 27 NgIoh5
- 21 NgIoh6
- 16 Shiwa

- 14 NGOpt
- 12 HyperOpt
- 11 SQPCMA
- 11 QODE

## F   Additional experimental figures for artificial problems in Nevergrad

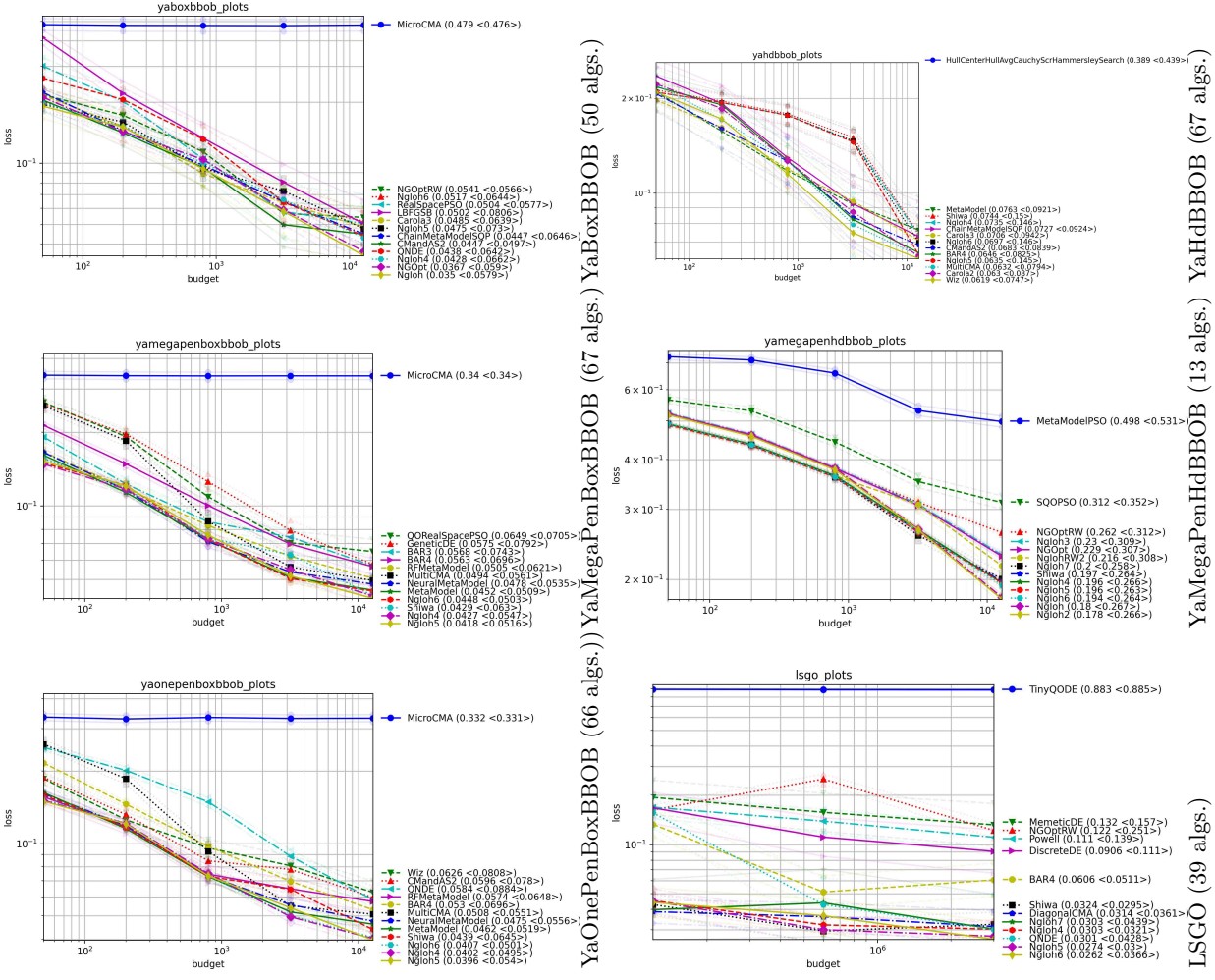

Figure 6: Variants of YABBOB with small ratio budget/dimension and LSGO. Other variants of BBOB in Figure 7. This is the average normalized loss (see details in Section 3.5), with only the best methods (NgIoh4 is always there) and the single worst; see Figures 9 to 11 for more methods in the frequency of winning figures. Right: name of benchmark and number of algorithms run.

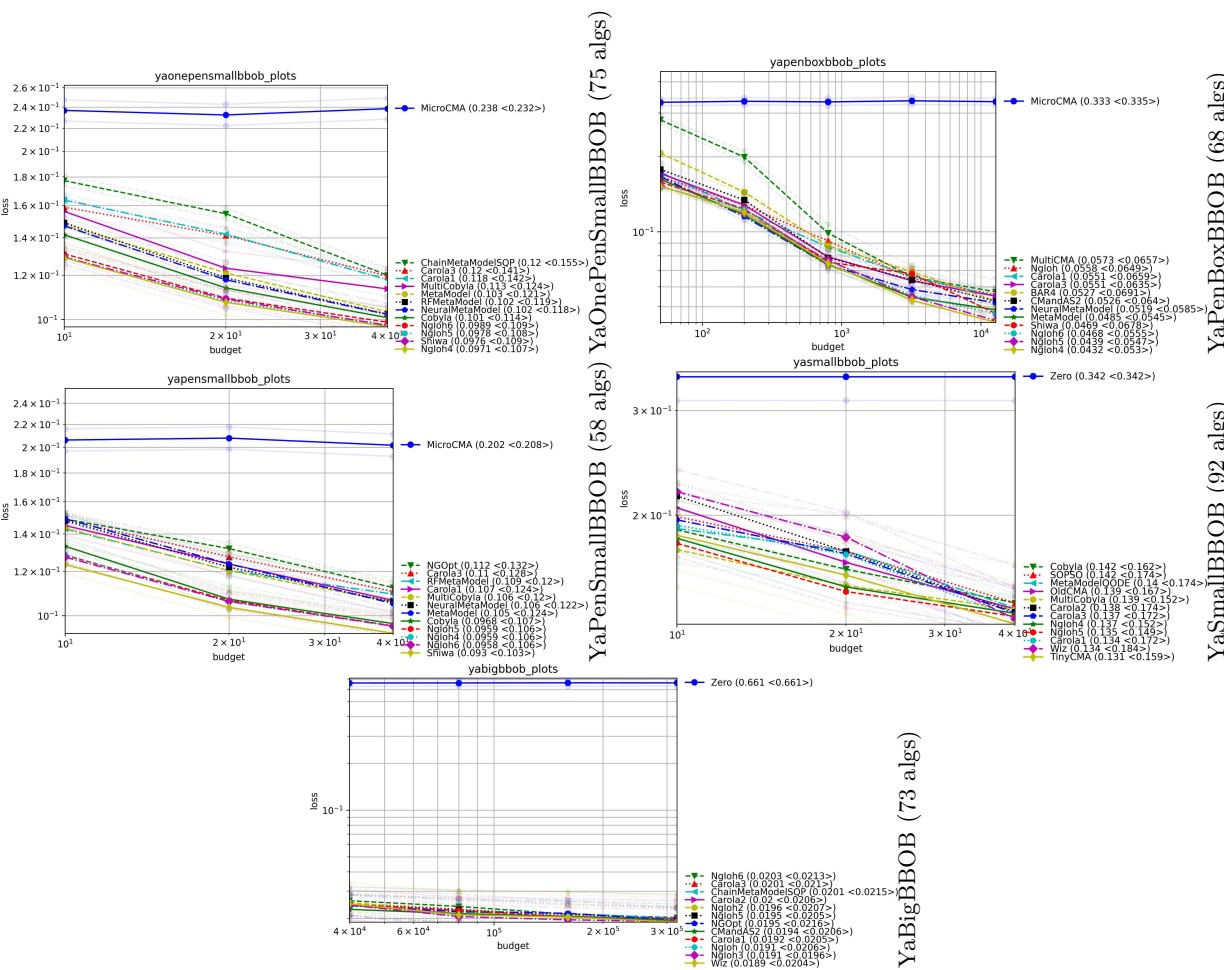

Figure 7: Variants of YABBOB with small ratio budget/dimension. The last one, YaBigBBOB, is the opposite, with a large ratio budget/dimension. Only the 12 best methods and the worst are presented, all benchmarks include several variants of CMA, DE, PSO and others (see referenced URLs or Fig. 11 for all details and more algorithms). Overall, NgIoh variants are excellent.

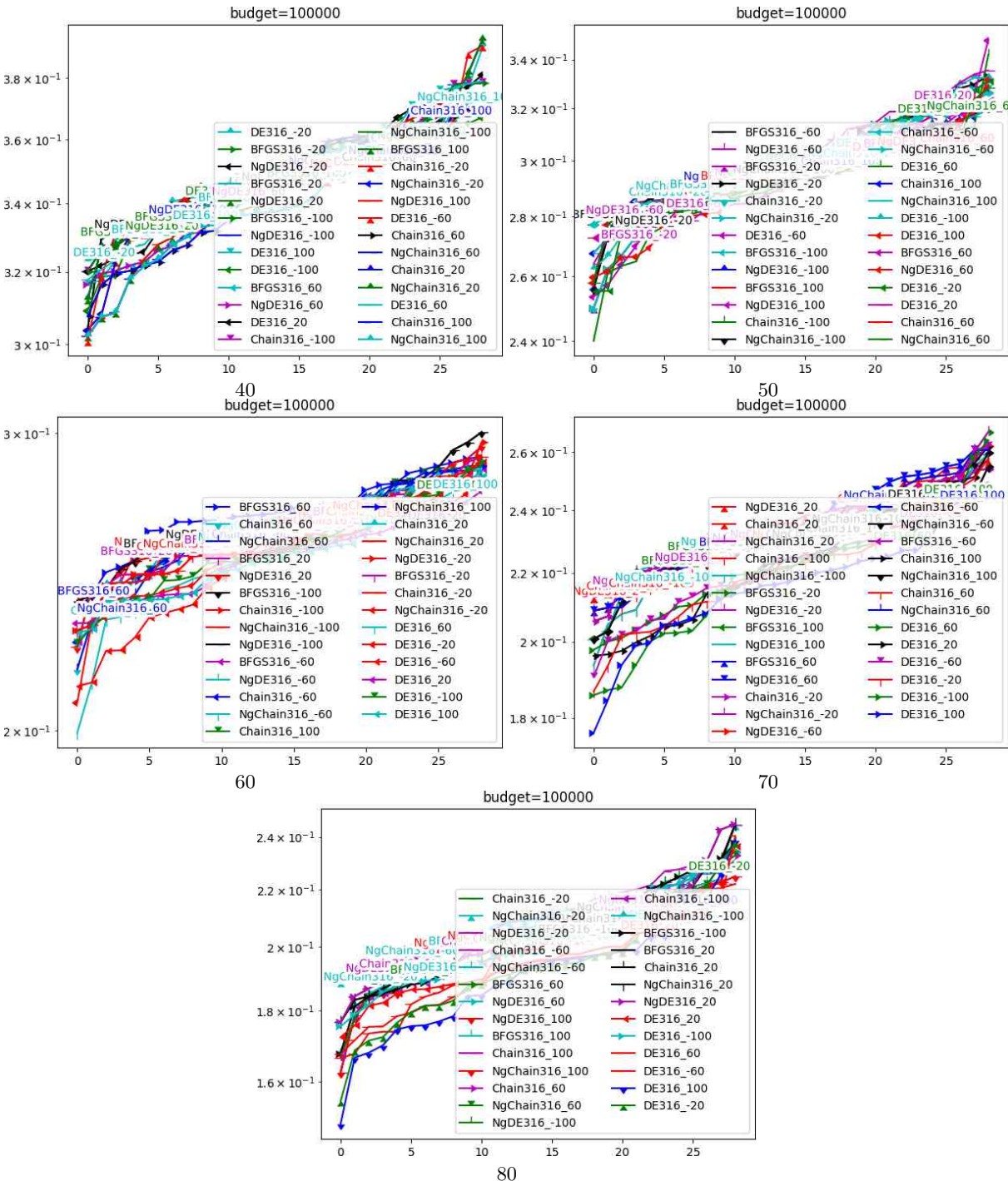

Figure 8: Photonics optimization (mirrors for various wavelengths) for 40, 50, 60, 70, 80 layers respectively: sorted result of the 30 runs of each method (best run on the left and worst run on the right). The 27 best (for the median) are presented (best at the end, right column, bottom), this extends Figure 14. For moderate numbers of layers, comparisons are unclear, whereas for large-scale versions DE dominates.

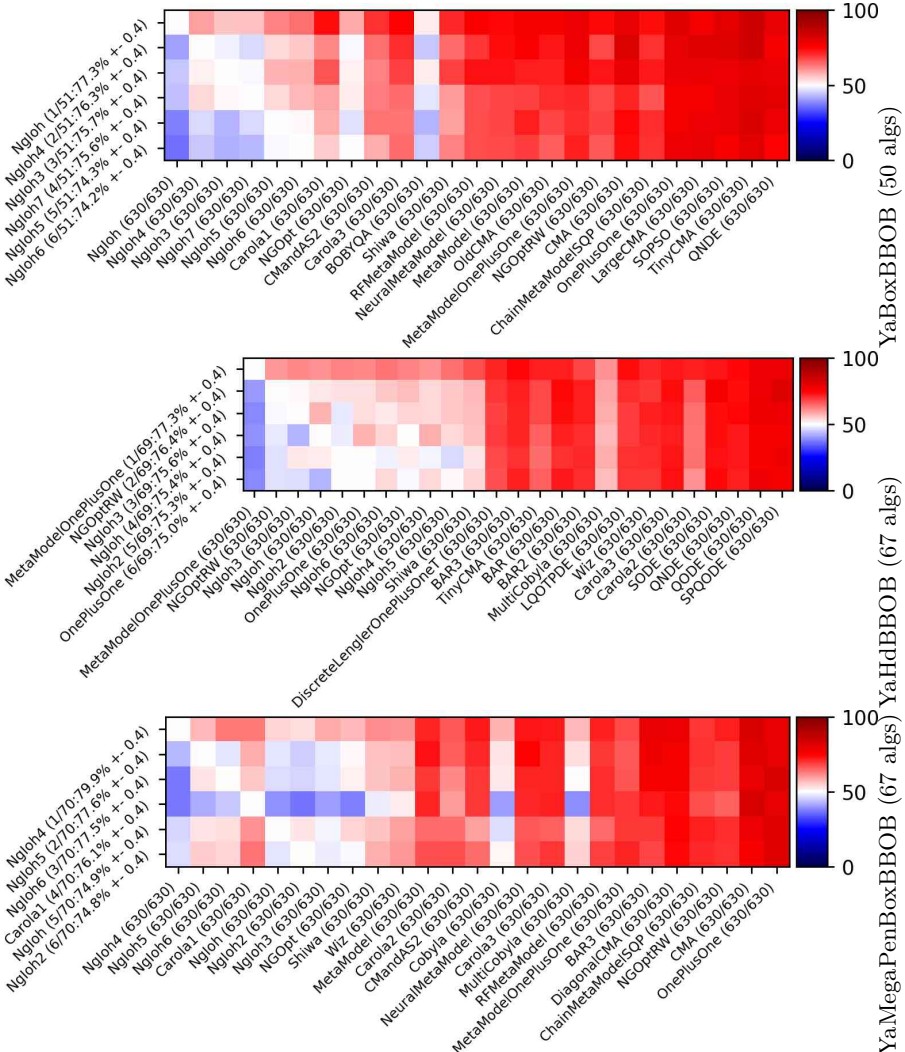

Figure 9: Variants of YABBOB with small ratio budget/dimension. Other variants of YABBOB in Figures 10 and 11. This is the frequency of winning figure (see details in Section 3.5, with the best methods on the left. NgIoh variants dominate.

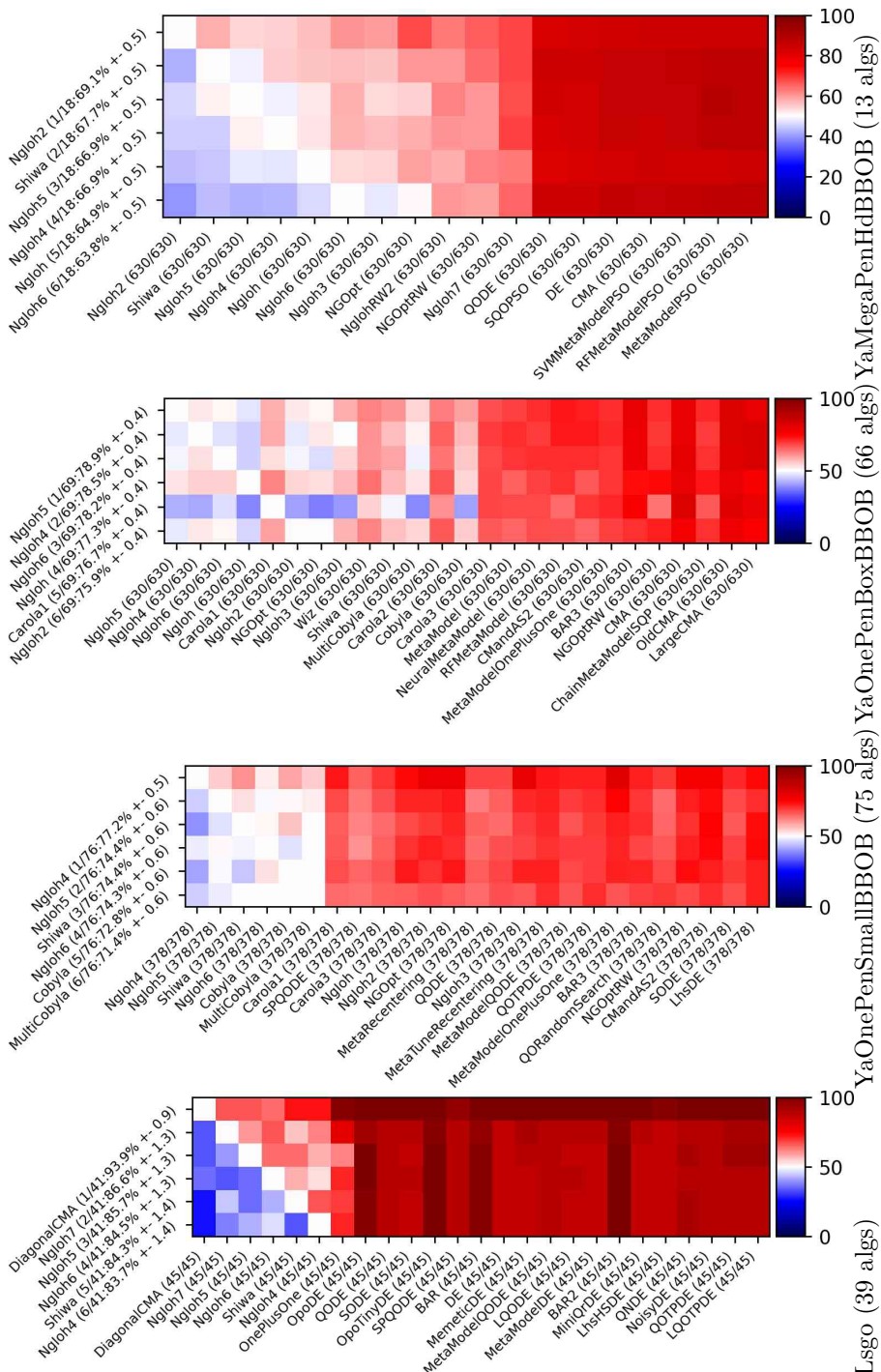

Figure 10: Variants of YABBOB with small ratio budget/dimension and LSGO (Li et al., 2013). Other variants of YABBOB in Fig. 9 and 11. This is the frequency of winning figure (see details in Section 3.5, with the best methods on the left. Total number of methods run on the right.

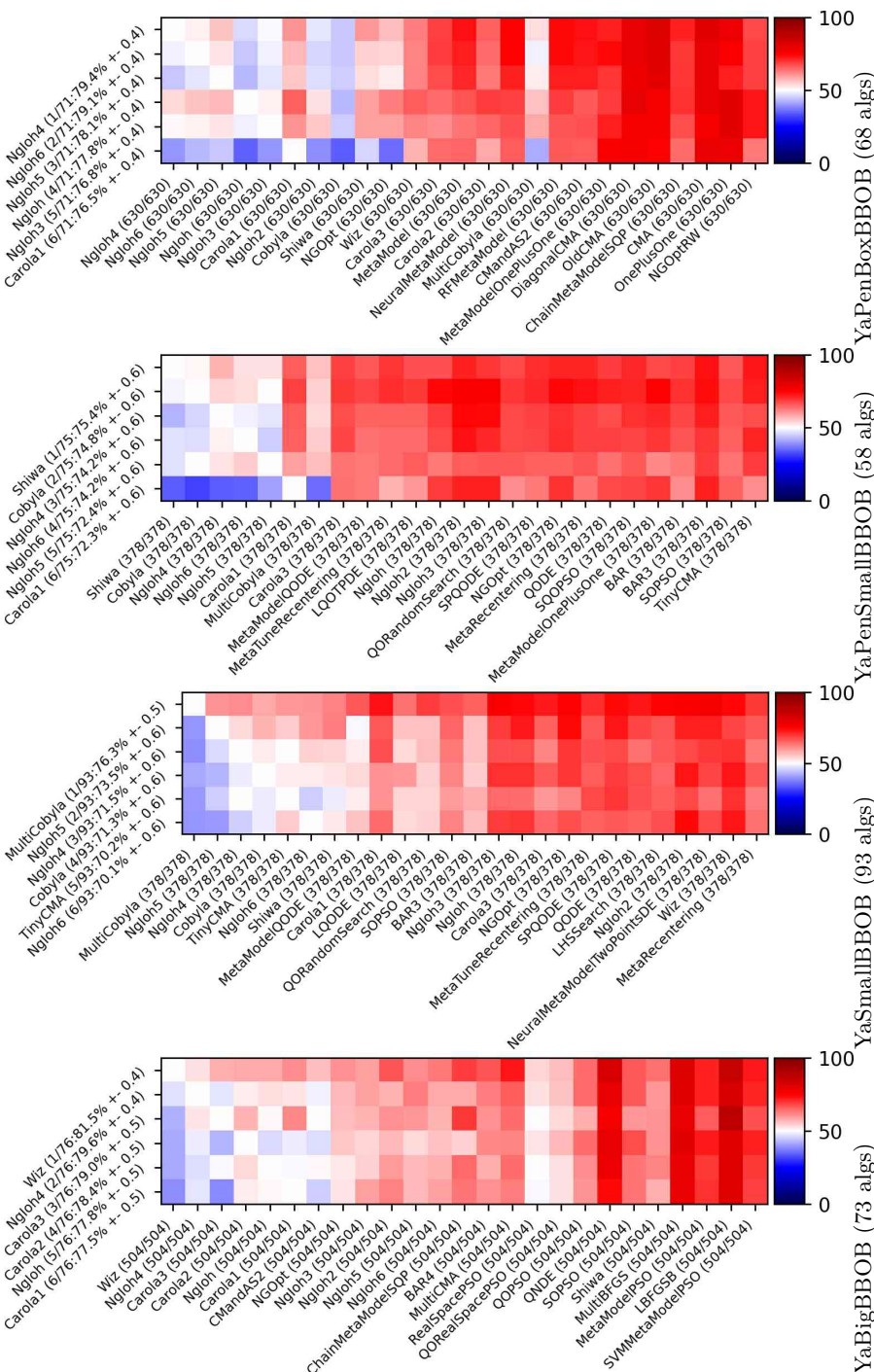

Figure 11: Top: 3 variants of YABBOB with small ratio budget/dimension. Bottom: YaBigBBOB, is the opposite, with a large ratio budget/dimension. Frequency of winning figure as detailed in Section 3.5, with the best methods on the left. NgIoh variants and (except for the last) Cobyla dominate.

# G  Outside Nevergrad: application to external real-world problems

Finally, we include a few use cases by Nevergrad users. The benchmarks and setups have been developed independently of the benchmarking platform included in Nevergrad. The plotting tools, functions, and criteria, are frequently different from the rest of the paper. They, on purpose, quantify the robustness of the conclusions drawn on our update of the Nevergrad benchmark, specifically for the real-world cases. Overall, results in Appendices G.1, G.2 and G.4 confirm the conclusion, in Nevergrad benchmarks, that DE performs well on many real-world problems; the discrete problem in Appendix G.3 confirms the good performance of Lengler though FastGA (Doerr et al., 2017) is also good; Appendix G.5 confirms the performance of SQOPSO when the scale of the optimum is unknown, in particular in the neuro-control case.

## G.1  Infrastructure: optimizing a caching policy

In this application, Nevergrad is used to optimize a caching strategy. The problem comprises 84 decision variables for the optimization. These variables encode the cache strategy. We run each method in several variants, with random parameters $a$, $b$, and $c$ so that constraints are penalized by $a \times constraintViolation^b \times i^c$, with $i$ being the iteration index. With this dynamical constraint penalization scheme, constraints violations are increasingly penalized so that eventually solutions without any violations are found. Compared to artificial benchmarks above, the setting has been influenced by the computational cost. All methods including GeneticDE, PSO, DE, TwoPointsDE, DiagonalCMA were run the same number of times, and the 11 best results are presented. We observe (Figure 15, left) that GeneticDE performs best and in general, DE variants perform well. One of the conclusions from this experiment is how much most Bayesian methods cannot be used for large budgets and dimension spaces larger than 84 (none of the methods available in Nevergrad was usable here), and computing gradients by finite differences (introducing a factor 85 in the computational cost) is also unfeasible. The results are consistent with the effectiveness, in our benchmarking suite, of DE variants for real-world problems with similar size/budget (Section 4.2). However, we would not have guessed the good performance of GeneticDE for this specific problem. Another observation is that we get a strong improvement compared to the handcrafted heuristic implemented before using the standard algorithm (+70%) and also better than the manually designed solution (initial point). The problem is repeated: there are frequently new versions to be solved, so that doing this experiment is useful for doing a choice of algorithm for the future. We note (unpresented experiments) that Inoculation (Inoculation, here, consists in adding in the population 8 points obtained in previous optimization runs) roughly reduces the computational cost by a factor of five. We get roughly the same performance with 20% of the budget.

## G.2  Crop optimization

This application combines Nevergrad, PCSE (de Wit, 2021), and NASA data (Sparks, 2018) for optimizing the choice of crops in many countries. Figure 12 presents a specialization of the code to Kenya, including choosing crops and their varieties, depending on climate. Compared to the original code in Nevergrad, there are additional variables, for choosing the crop and the variety. After the present performance check (confirming the good behaviour of NGOptRW), a forthcoming publication is under work for various crops and continents.

## G.3  Mobile Network Base Station Placement Optimization

Figure 13 presents the experimental results regarding the optimization of the placement of base stations for a mobile network. An original ad-hoc implementation already existed before testing Nevergrad on this problem. The method which typically performs best in our discrete benchmarks, namely Lengler (Doerr et al., 2019; Einarsson et al., 2019), which uses a fixed, predefined mutation schedule and FastGA (Doerr et al., 2017), which is also a method with a fixed mutation schedule, but here the schedule is a stationary stochastic random variable. We observe that while methods in Nevergrad perform well for low budget and outperform the original method by far, the original method performs best for greater budgets. Seemingly, the key point is that it uses specific mutation operators, whereas Nevergrad focuses on lists of variables with

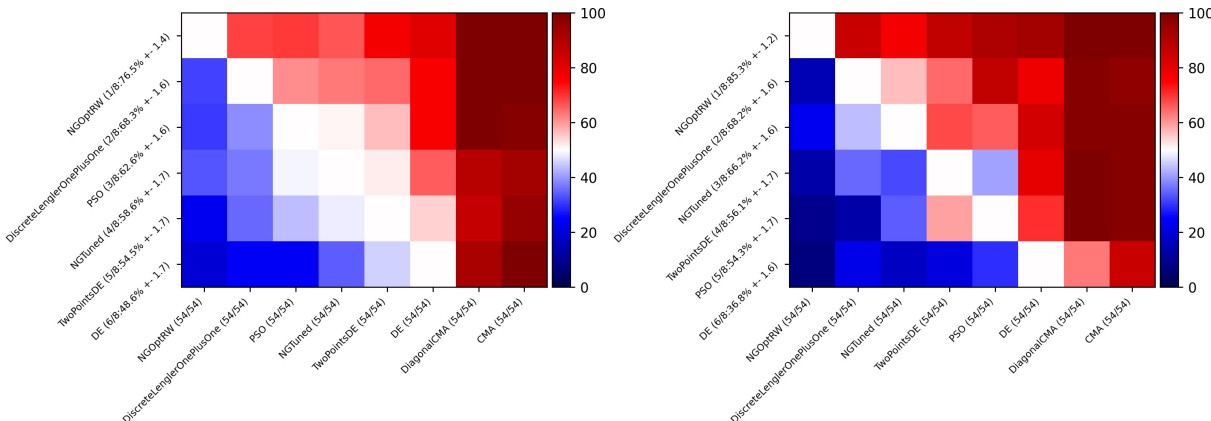

Figure 12: Comparison between optimization methods for crop optimization in the case of Kenya (left: 2011, corresponding to a particularly dry year; right: 2006). Setup as in Section 3.5: the heatmap shows the frequency at which method X (row) outperforms method Y (col). Rows and cols are ranked by average frequency against all other methods: top/left is best. As in many real-world cases, NGOptRW is excellent.

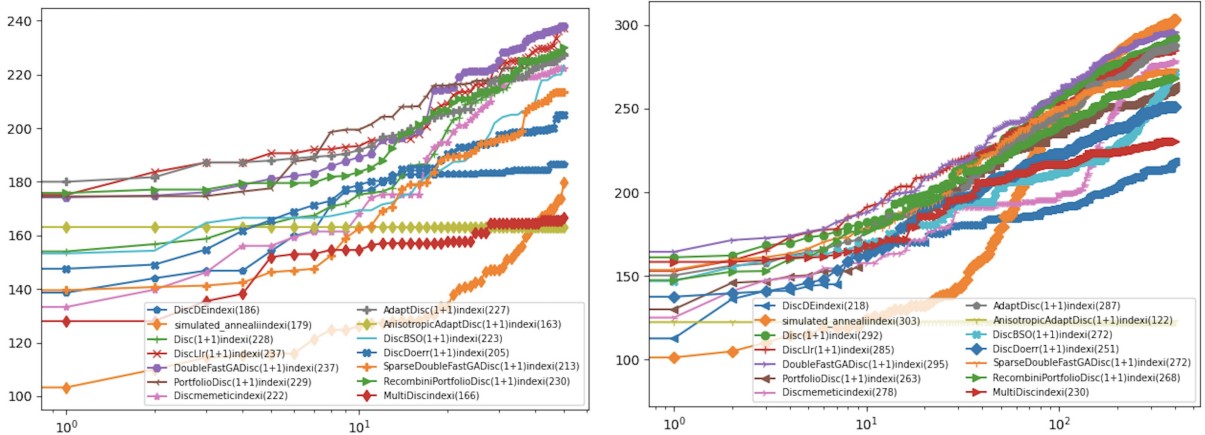

Figure 13: Placement of base stations of a mobile network: optimization with budget 50 (left) and 400 (right): the greater, the better, average best score between parentheses. We observe that Nevergrad methods performed quite well for the low budget case, but the specific method (Simulated Annealing with ad hoc mutation operator, in orange) developed for the problem at hand is the best for budget 400. Between parenthesis, the best obtained value. Llr is short for Lengler (Doerr et al., 2019; Einarsson et al., 2019).

generic operators. Nevergrad improves results on this 200-dimensional problem when the budget is 50, but not with budget 400.

## G.4    Robust topology optimization

Figure 14 presents the results for the optimization of mirrors smaller than a micron aimed at reflecting light at wavelength between 400nm and 650nm using only two materials.

Only the 7 best performing methods are presented, but actually 30 methods are run: There are 5 algorithms: DE, BFGS, Chain (a chaining of DE during half budget, followed by BFGS) from Nevergrad and the DE and Chain from Pymoosh (Langevin et al., 2023). For differentiating methods from Nevergrad, we add a prefix Ng for those methods. Each of them is run with a sampling parameter in $\{-100, -60, -20, 20, 60, 100\}$, hence 30 methods. This parameter specifies how robustness to wavelength is taken into account and has little impact here. The detailed description is beyond the scope of this paper. Another sampling parameter is

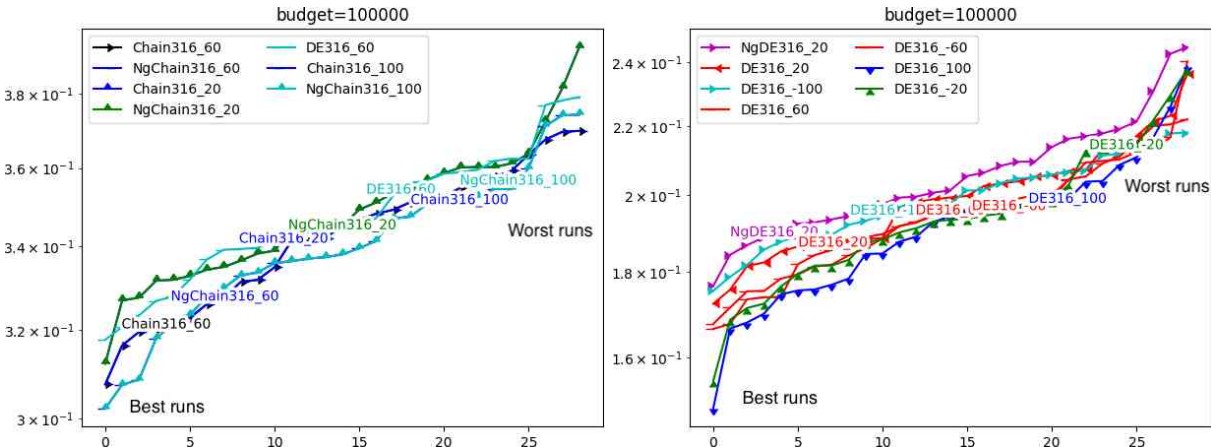

Figure 14: We perform photonics optimization (mirrors for various wavelengths) for 40, 50, 60, 70, 80 layers respectively: we keep only 40 and 80 for short and refer to the appendix for more. For each algorithm, we plot the results of 30 runs (best on the left, worst on the right): a horizontal curve means constant results, whereas a sharp increase means variable results. Methods are ranked by median value. Only the few best are presented for readability (legend: best at the end, i.e. bottom of the right column): extended version with more algorithms in the ranking in appendix, Figure 8. For moderate numbers of layers, the ranking is unclear (with Chaining of DE and BFGS frequently good), whereas for large-scale versions, DE dominates (for 80 layers, the 6 codes based on DE corresponding to the 6 values of the sampling parameter dominate all 24 other codes). The impact of the sampling parameter (suffix of the algorithm name) is unclear.

fixed at 316 (the square root of the budget) after preliminary experiments: it is actually the most important choice in the optimization design, other values are removed from plots as this is not the point in the present paper.

We observe that all strong methods, in the highest dimensional cases, are DE (either the one from Nevergrad, which is quite standard, or the one in PyMoosh which has been optimized for the problem at hand). This limited comparison validates the choice of DE in PyMoosh, though testing more algorithms could be possible. In lower dimension, we observe that the chaining of DE and BFGS frequently performs better than DE or BFGS alone.

### G.5 Gym

Nevergrad contains OpenAI Gym problems, which were deprecated after the issues of Gym v0.24.0, so that Gym was not included in recent exports of the Nevergrad benchmarks. We update the code importing Gym and rerun the experiments. Our code is merged in the codebase. Results are presented in Figure 15 (right): SQOPSO performs well.

## H    Additional results with more variants

Since the first version of the present paper, several new variants of NGOpt were added (by us) in the Nevergrad platform, including

- variants of NgOpt based on new direct search methods in (Roberts & Royer, 2023), with NgDS in the name. These variants derive from our NgIoh but use the stochastic direct search from Roberts & Royer (2023) instead of CMA in sequential cases.

- variants of NgOpt based on LogNormal mutations, such as NgLn, which are based on NgIoh but with Cobyla replaced by LogNormal mutations Kruisselbrink et al. (2011b) as a first step.

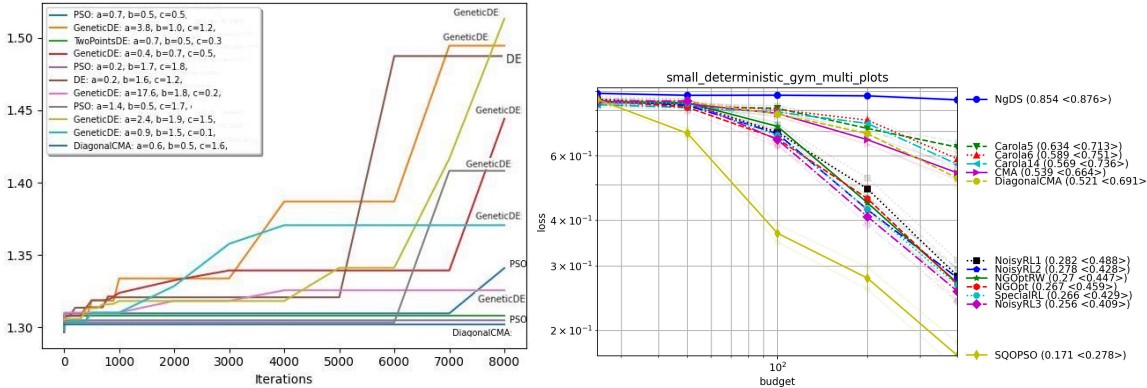

Figure 15: Left: Comparison of various methods on the infrastructure problem. The upper the better, only the 11 best results are presented: GeneticDE is frequently among the 11 best, whereas all methods were run the same number of times. Right: Experiments on Gym (more algorithms and more benchmarks in Section 19), confirming the good performance of SQOPSO (and existing wizards dedicated to reinforcement learning, with RL in the name Rapin & Teytaud (2018)) for neurocontrol.

The present appendix presents results comparing our algorithms SQOPSO and NgIoh4, algorithms above, and two more algorithms based on our proposals NgIoh and SQOPSO:

- NgIoh21 is NgIoh4 with constants within Carola2 modified (10% for Cobyla, 80% for the CMA with MetaModel, 10% for the final convergence with SQP).

- NgIohTuned (available in Rapin & Teytaud (2018)) is similar to NgIoh21 but it uses (i) NgDS instead of NGOpt in sequential cases (ii) VLPCMA (i.e. CMA with population size multiplied by 100) instead of CMA when the budget is greater than 2000 times the dimension (iii) most importantly, additional information provided by the user (if any) for switching to different algorithms: mostly, it switches to SQOPSO for neural control and to NGOptRW for other real-world problems.

- SQOPSODCMA, a chaining of SQOPSO (half budget) and Diagonal CMA (second half of the budget).

Basically we observe on results below that NGOptRW (using a lot of DE and PSO) is preferable to NGOpt in many real-world contexts, that SQOPSO is better in the neuro-control case, and NGOpt variants using a first exploration step by Cobyla or other methods (as our method NgIoh4 does) perform better than NGOpt in particular for benchmarks such as MS-BBOB and ZP-MS-BBOB which carefully control for the norm of the optimum.

## H.1 Additional results: YABBOB, ZP-MS-BBOB, MS-BBOB

We observe in Fig. 16 and 3 that results on YABBOB are mixed, whereas for MS-BBOB and ZP-MSBBOB all strong methods use either Cobyla or quasi-opposite sampling as a first step, validating our contributions:

- MS-BBOB: NgIoh4, SQOPSODCMA, NgIohTuned, NgIoh21, SQOPSO, all outperform all other methods, including NGOpt.

- ZP-MS-BBOB: NGIoh4, SQOPSODCMA, NgIoh21, NgIohTuned, all outperform all other methods, including NGOpt.

We note that the succesful codes (outperforming NGOpt) are exactly

- the codes using quasi-opposite sampling as a first stage (as SQOPSODCMA, in both benchmarks) before using a local convergence method;

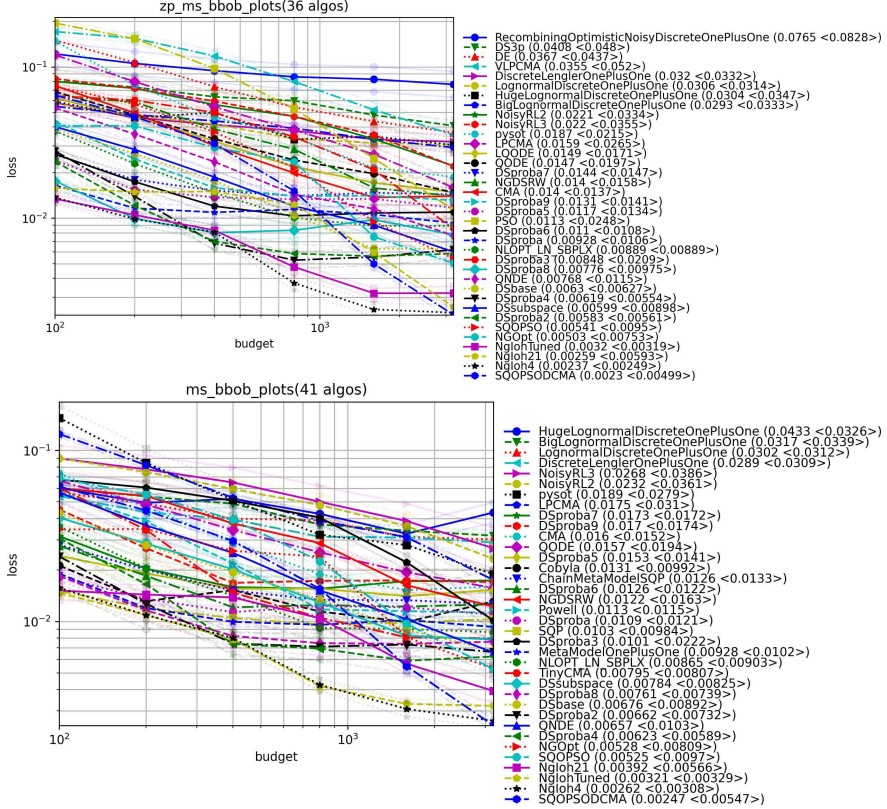

Figure 16: Additional results on ZP-MS-BBOB, MS-BBOB with more algorithms (see Fig. 3 for the case of YABBOB, i.e. without the multiscale approach of ZP-MS-BBOB and MS-BBOB). NGOpt is competitive on YABBOB (and our methods are not always at the top, though they are good), but fail (and many methods as well) compared to our methods in the cases of MS-BBOB and ZP-MS-BBOB. This shows how much results are different when we consider multiple scales. Note that besides the good performance of NgIoh4 and NgIoh21 and NgIohTuned on ZP-MS-BBOB and MS-BBOB for the greatest budgets, the curves are also lower than the NGOpt curve for the various budgets.

- and the codes which use our chaining initiated by Cobyla (NgIoh variants);

except, in one of the two benchmarks only, SQOPSO, which still has quasi-opposite sampling but no second stage. By contrast, all other codes, except SQOPSO for one of the two benchmarks, perform worse than NGOpt. These results confirm the relevance of quasi-opposite sampling or Cobyla as a first stage for problems equipped with multiple-scale such as MS-BBOB or ZP-MS-BBOB.

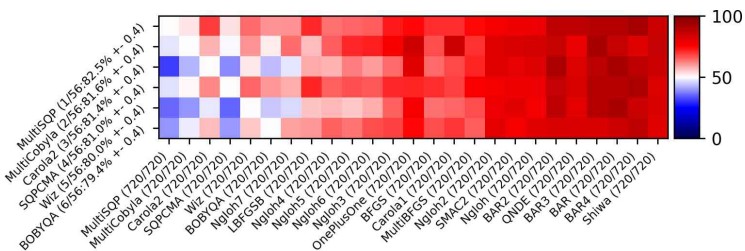

Figure 17: Counterpart of Fig. 2 for ZP-MS-BBOB with the frequency of winning (26 best, out of 58 methods) instead of the normalized average loss. This takes into account all budgets as detailed in Section 3.5, hence the ranking is not the same as in Fig. 2 which ranks only based on the results for the maximum budget, We note that many mathematical programming methods (using SQP, Broyden–Fletcher–Goldfarb–Shanno (BFGS) with finite differences, BOBYQA or Cobyla) are excellent for low budget (so that they get a good ranking on the right), which corroborates the idea of using Cobyla as a first step in Carola2. 58 algorithms are run (the 26 best are presented), and the previous wizard, NGOpt, is ranked 22 and CMA is ranked 38 (this differs from statistics on normalized average loss as in Fig. 2 but we still observe a superiority of NgIoh methods compared to CMA or NGOpt, a conclusion which is not so clear on e.g. YABBOB).

## H.2 Additional results: real-world problems from Nevergrad

Figure 18 presents results of many algorithms on Aquacrop. We observe that on this real-world problem, all successful algorithms use DE or PSO, confirming the excellent behaviour of these algorithms in such settings. Also, the top algorithms frequently use either quasi-opposite sampling or GeneticDE (as in NoisyRL3, a wizard specifically designed for reinforcement learning), both aimed at taking care of the scale (NoisyRL3 and NGDSRW use GeneticDE, LQODE is a DE with quasi-opposite sampling, SQOPSODCMA uses quasi-opposite samping).

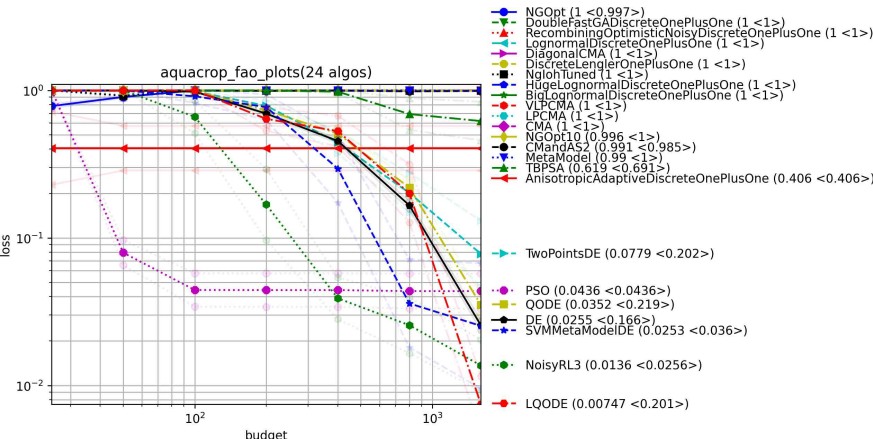

Figure 18: NGOpt is outperformed on the Aquacrop problems by all algorithms based on DE or PSO, even more when these algorithms use quasi-opposite sampling.

Fig. 19 confirms the good performance of SQOPSO for neural control in deterministic contexts, with the application to Gym.

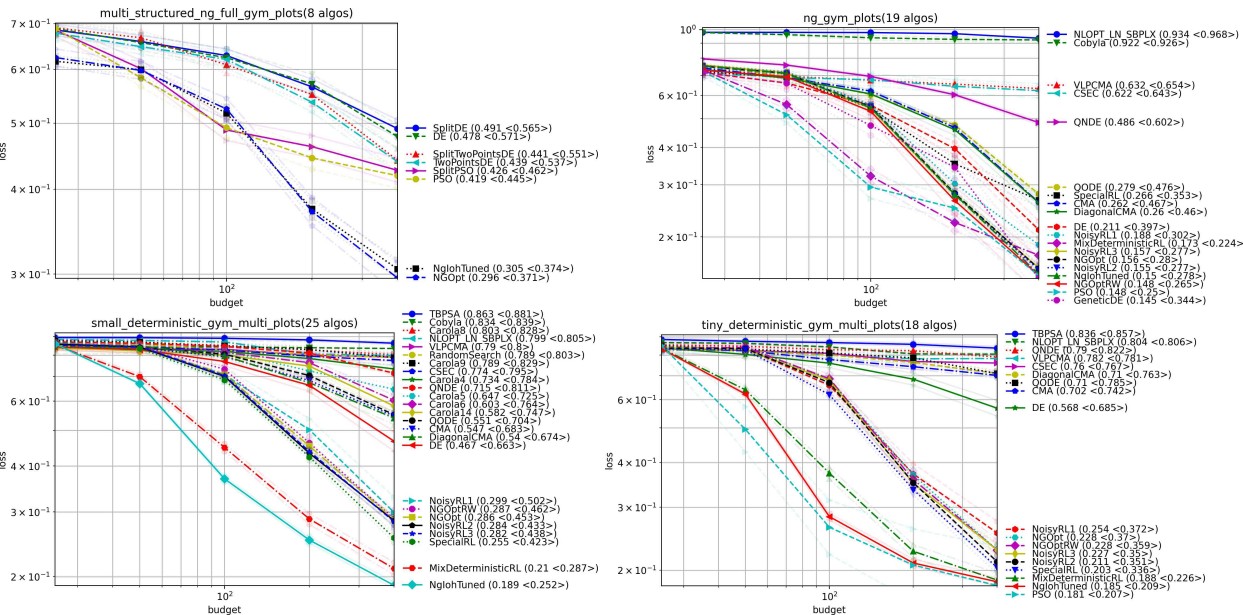

Figure 19: In the case of neural control, NgIohTuned (which uses the knowledge that this is a neural problem) switches to SQOPSO, hence its excellent performance. This figure confirms the excellent performance of SQOPSO for neural control.

### H.3 Additional results comparing NgIoh to NGOpt on the old BBOB/COCO

We compare

- NgIoh4,

- NgIoh21 which is a more recent version of NgIoh4 with a tuning of constants of the chaining (10%, 80% and 10% instead of three thirds in Carola2) and increasing the population of CMA by a factor 100 (compared to the default) for budget greater than 2000 times the dimension,

- NgIohTune, a more sophisticated improvement based on NgIoh21, but also high level information such as "is a real-world problem" or "is a neuro-control problem" for selecting NGOptRW and SQOPSO respectively.

- NGOpt,

- b-cmafmin2 (the baseline CMA included in the code of BBOB/COCO),

- u-CmaFmin2 and r-CmaFmin2 correspond to the same CMA but with different restart schemes (the same restart as for Scipy methods for rCma and uniform random restarts for rCma, whereas CmaFmin2 uses a proposal function "propose-x0" from the objective in Coco/Bbob), and other variants of CMA found in Nevergrad.

We have 24 cases, corresponding to dimensions 2, 3, 5, 10, 20, 40 and budget/dimension 10, 100, 1000, 10000. NgIoh4 and NgIoh21 both outperformed NGOpt in 19/24 cases, NgIohTuned outperformed NGOpt in 21/24 cases. NgIoh4 outperforms bCmaFmin2 in 14/24 cases. NgIoh21 outperforms bCmaFmin2 in 17/24 cases. NgIohTuned outperforms bCmaFmin2 in 19/24 cases and is frequently the best overall. Overall, NgIoh4 and all its variants outperform NGOpt on BBOB/COCO.

Results in low budget cases confirm the excellence of Cobyla already observed in Dufossé & Atamna (2022); Raponi et al. (2023).

### H.3.1 Additional results: BBOB with budget = 10× dimension

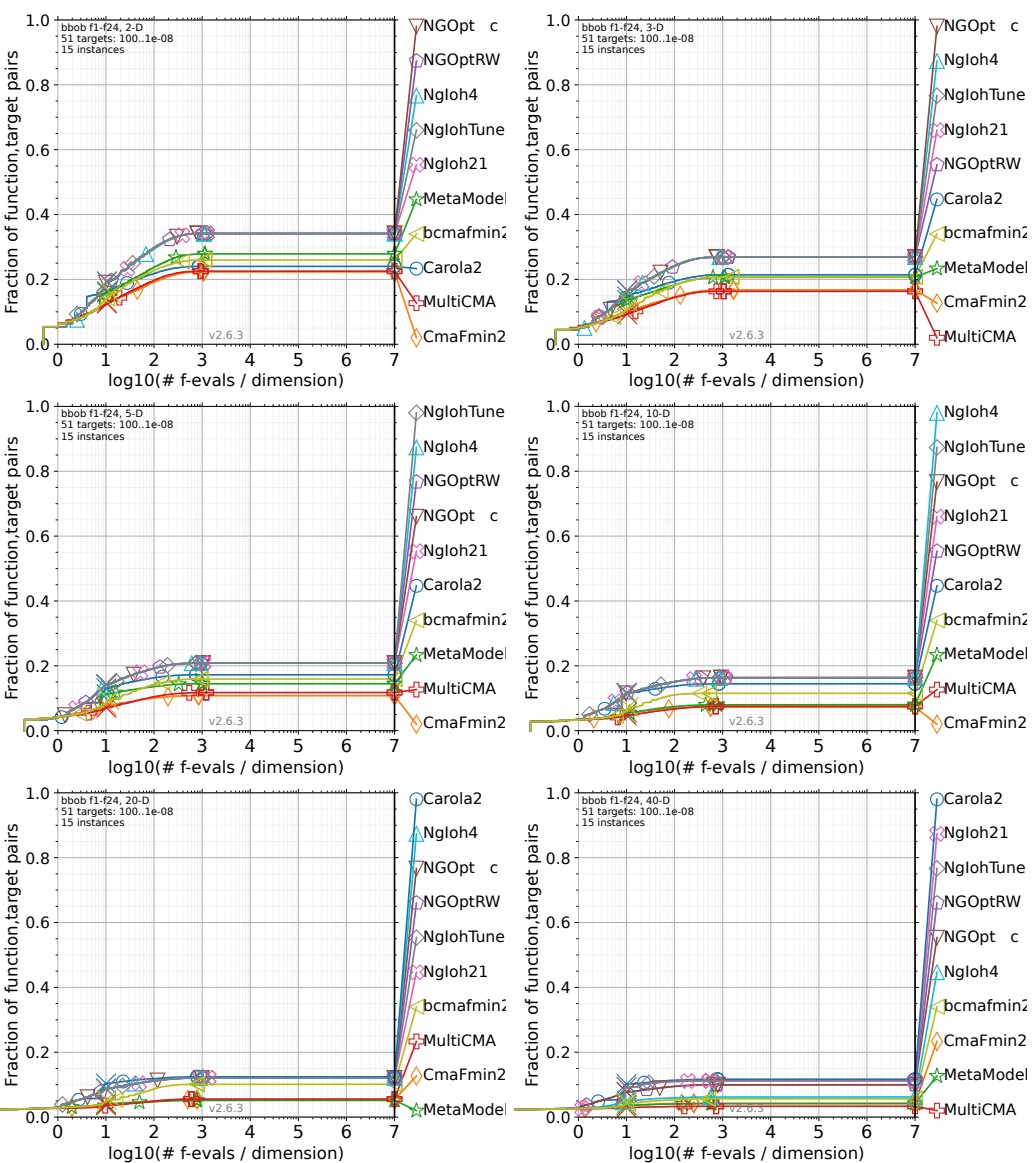

Figure 20: BBOB with budget $10\times$ dimension. The higher the better on these BBOB/COCO figures.

## H.3.2   Additional results: BBOB with budget $= 100\times$ dimension

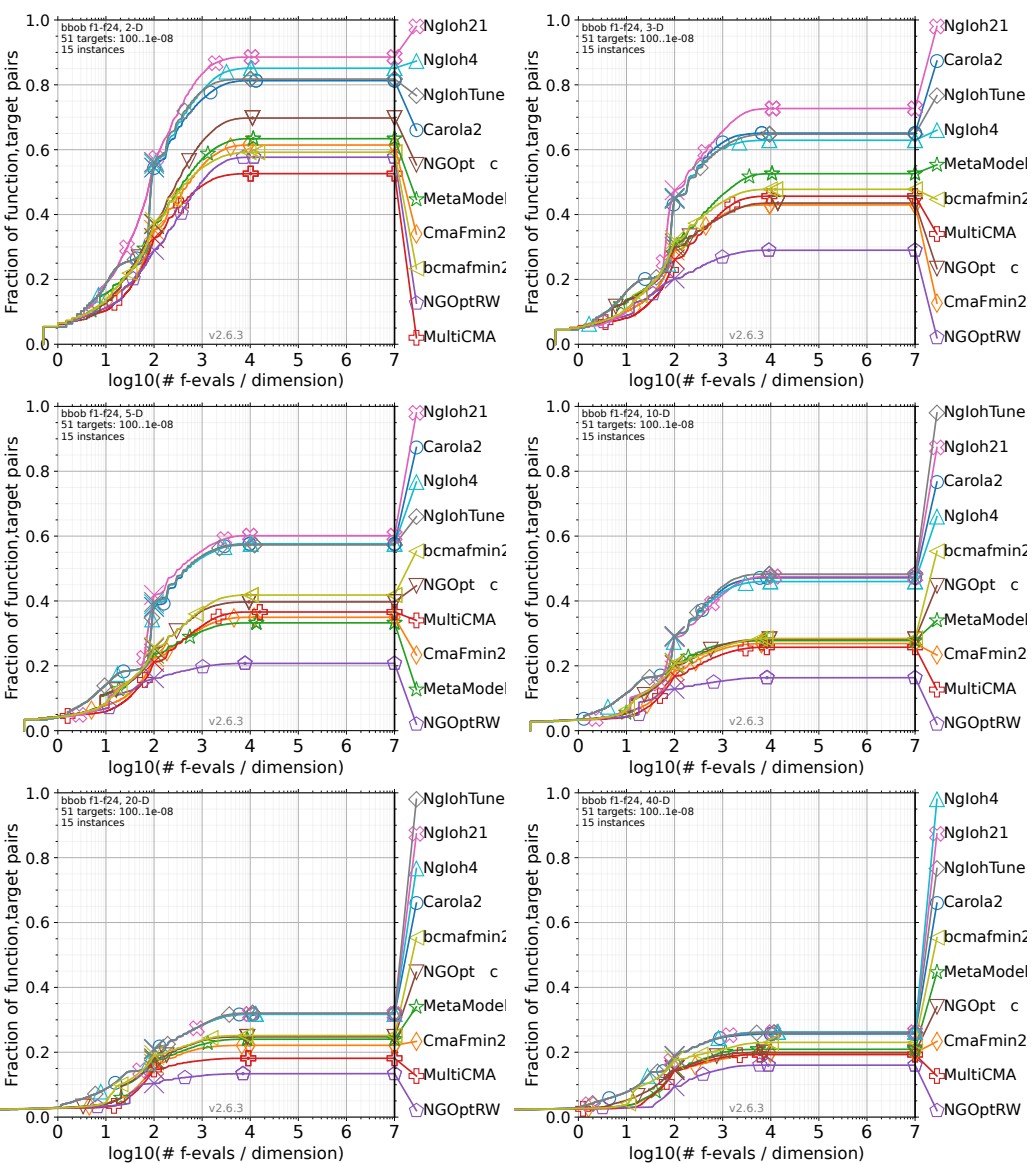

Figure 21: BBOB with budget $100\times$ dimension. The higher the better on these BBOB/COCO figures.

### H.3.3 Additional results: BBOB with budget $= 1000\times$ dimension

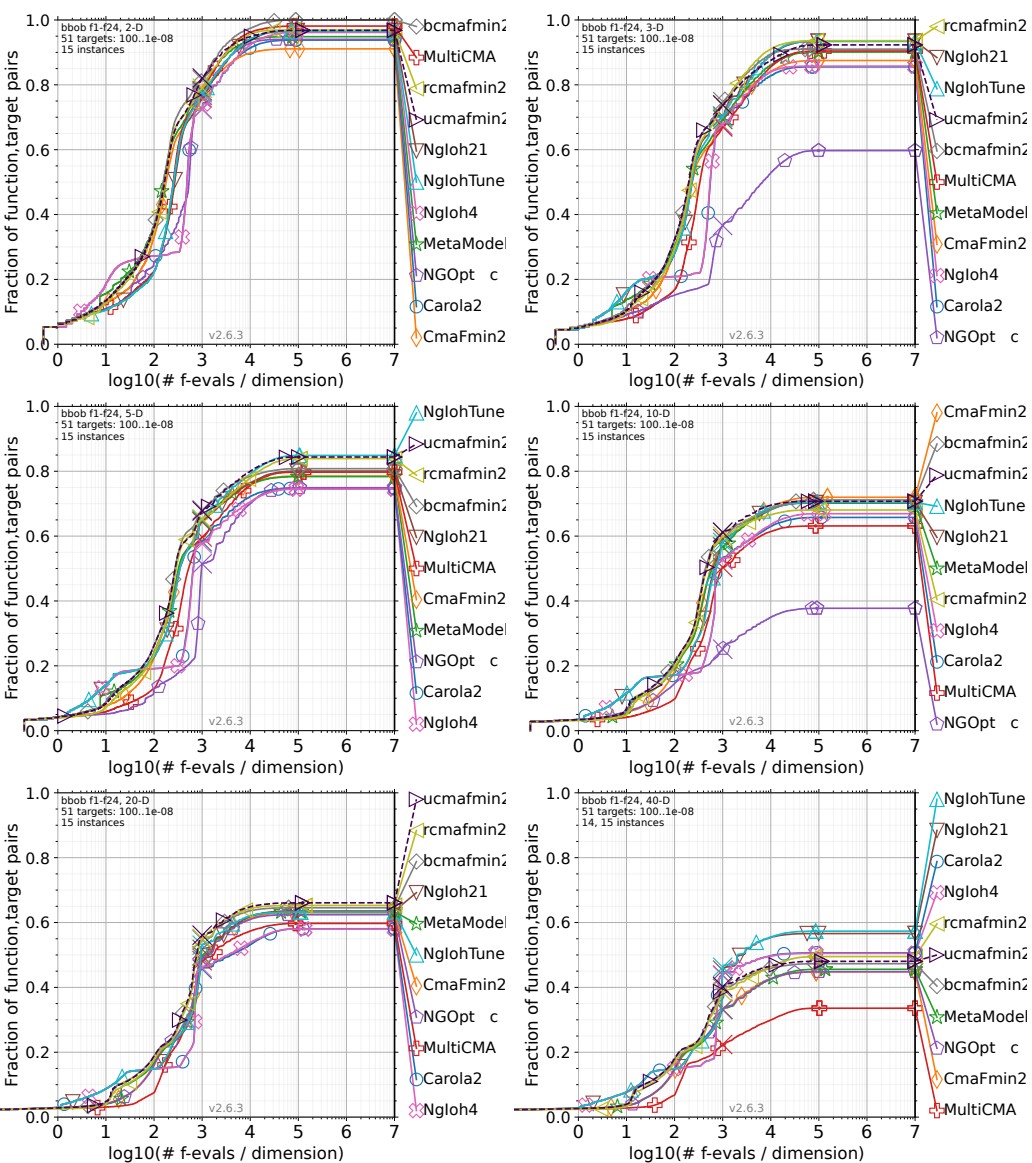

Figure 22: BBOB with budget 1000× dimension. The higher the better on these BBOB/COCO figures.

## H.3.4   Additional results: BBOB with budget $= 10000\times$ dimension

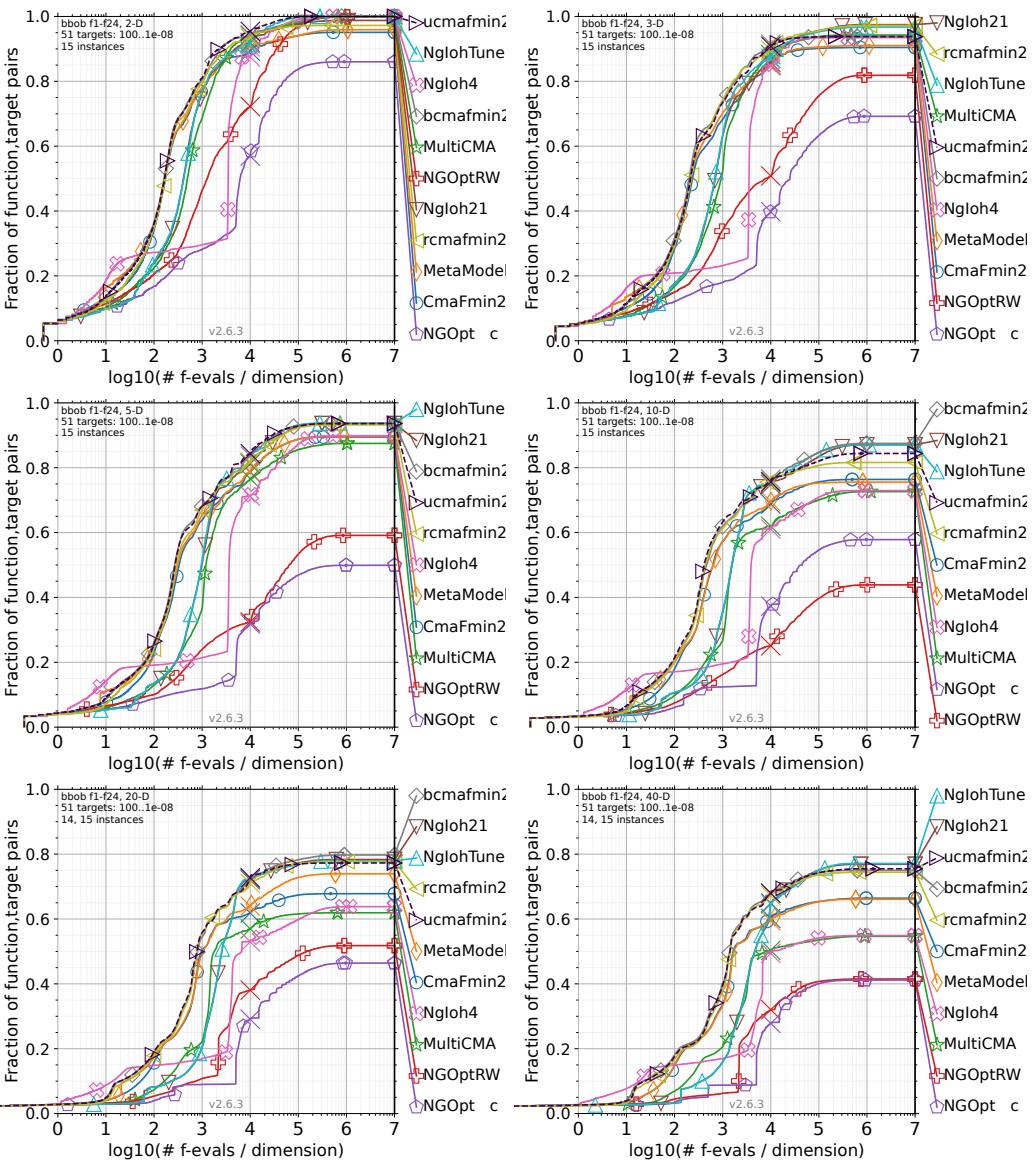

Figure 23: BBOB with budget 10000× dimension. The higher the better on these BBOB/COCO figures.

