# OpenReview forum: "NgIohTuned, a new Black-Box Optimization Wizard for Real World Machine Learning"
_TMLR — Rejected by TMLR_

### Review · Reviewer_1gfJ · 2024-03-11

**Summary Of Contributions:**

The paper proposes a set of new benchmarks for black-box optimization that
consider the distance of the optimal point from the optimum, evaluates them
empirically, and a set of black-box optimization algorithms that leverage the
new benchmark set and empirical observations.

**Audience:**

Yes

**Broader Impact Concerns:**

No concerns.

**Claims And Evidence:**

Yes

**Requested Changes:**

No changes requested.

**Strengths And Weaknesses:**

The paper is well-written, presents interesting results and an interesting
perspective on black-box optimization problems. My only complaint is that the
paper is very dense and covers a lot of ground. I feel that a longer paper (or
reduced scope) would do the material more justice. Overall, this is a very nice
paper though!

The anonymity of the authors is somewhat compromised by the naming of the
algorithms...

---

> ### Author Response · Authors · 2024-03-26
> **We extend the paper for clarifying (in particular, we add a big appendix).**
>
> Thank you for your review and comments. We agree that our tables of results are too aggregated, and that searching the big online archive of data is tedious. Therefore we add an appendix, including in particular more figures about YABBOB, MS-BBOB, ZP-MS-BBOB (which are helpful for understanding the key point of scaling) and BBOB/COCO and GYM, with the current version of Nevergrad (including several new methods). The key point is that all strong methods, when including ZP and/or MS, are based on an initial search by Cobyla or by Quasi-Opposite sampling. Likewise, the effectiveness of NGOptRW (which uses DE and PSO) is confirmed by additional real-world results. We feel that these additional results clarify the importance of Cobyla or Quasi-Opposite sampling, and of DE/PSO for real-world black-box optimization. Also, for clarifying the paper which is too dense, we added a long appendix including NgIohTuned; whereas Carola2 and NgIoh4 was illustrating mostly the problem of scale, NgIohTuned (using high-level information about the problem, such as “is real-world” and “neuro-control”) incorporates all our conclusions and, as a consequence, as a clearer performance gap compared to other methods including the previous wizards in Nevergrad such as NGOpt.
>
> As mentioned, the risk of anonymity issue is low: the naming is close to the first name of an author of a cited paper, not of an author of the present paper.

---

### Review · Reviewer_gVy2 · 2024-03-13

**Summary Of Contributions:**

This paper addresses black-box optimization with (mainly) limited function evaluation budgets. The main contributions of this paper are twofold.

The first one is the modification of the benchmarking problems. Referencing Meunier et al. (2022), the authors claim the significance of the distribution of the optima of the problem. In particular, it matters whether it is close to the origin (I suppose the authors mean the center of the search domain) or to the boundary of the search domain. To make the benchmarking problems have varying distribution of the optima in the testbed, the authors introduces a scaling factor for the variables. The authors claim in the introduction that “we observe results in these problems with increased diversity of the distribution of the optima similar to those observed in real-world benchmarks.”

The second one is the novel algorithmic components to tackle the difficulty of unknown scaling introduced above. The authors claim in the introduction that “it improves the state of the art on average on many benchmarks.”

**Audience:**

Yes

**Claims And Evidence:**

No

**Requested Changes:**

Please refer to the weaknesses listed above.

**Strengths And Weaknesses:**

Strengths

1. A very simple modification that controls the distribution of the optima of the problem in the search domain is proposed. The proposed modification of the benchmarking problems, i.e., introduction of the scaling factor, would be helpful to know whether the tested algorithm is robust against the change of the scale.

2. The benchmarking platform including the proposed modifications is publicly available.

3. Promising performance of the proposed search strategy is shown in the experiments.

Weaknesses

1. The difficulty of solving problems with the optima located near the boundary comes from two factors: constraint handling (e.g., inadequate penalization would create a skew-landscape as in the skew-rastrigin function) and distance from the initial distribution (e.g., approaches without the mechanism to control the overall variance of the search distribution as in the step-size control mechanism in evolution strategies is slow to move and causes premature convergence). Though the proposed modification reveals the robustness against the change of the scale, it does not reveal why it is not robust when it is not robust. In particular, we do not know whether the tested algorithm employs an inadequate constraint handling or it is not good at controlling the overall variance. Without using the proposed framework, one can test these properties by simply testing an algorithm with/without boundary, and by using different initial search distributions or initial search intervals. Therefore, the usefulness of the proposed modification is rather limited.

2. The claim in the introduction “In particular, we observe results in these problems with increased diversity of the distribution of the optima similar to those observed in real-world benchmarks” is definitely an over-claim as the authors performed only a limited number of some real-world applications.

3. Usefulness of the proposed approach, Carola, is rather questionable. The reported comparisons are very limited. What differentiate Carola from other algorithm portfolios and when it is promising and when it is not? More comparison to other algorithm portfolios and investigation are required. No sufficient literature review is provided in the paper.

---

> ### Author Response · Authors · 2024-03-26
>
> Thanks for your review.
> 1. Thanks for this comment, we clarify the paper regarding this idea of “near the boundary”. Regarding constraints handling, here the problem is mostly not about constraints handling. With our scaling factor, we switch from cases in which the optimum is drawn in the entire domain with a norm of the same order as the distance between the middle and the boundary, and cases in which it is close to the middle. What we meant by “near the boundary” is the first case: this is not near the boundary as in cases in which the bound is active or very close to be active. In particular, no constraint is active at the optimum in YABBOB, MSBBOB and ZPMSBBOB. We apologize for not having been clearer in the first version and revise the paper accordingly: this is not related to constraint handling.
> 2. We include the 21 real-world benchmarks in Nevergrad (many of them including several distinct test cases), and 5 real-world benchmarks which were not in the version of Nevergrad before our work (Section 6). They support the importance of the scale, with the success of quasi-opposite samplings in Gym and in global statistics on real-world problems (Section 4.2). We add more appendices, including different views of the Gym problem which support the performance of SQOPSO in that case. We also add the results of NgIohTuned, which uses high-level information such as ``is a real world problem’’ or ``is a neuro-control problem‘’: NgIohTuned performs well on both artificial and real-world problems and aggregates all our conclusions in a single optimization wizard. We believe NgIohTuned clarifies the paper in the sense that it includes, in a single code, all conclusions from the paper, both for artificial and for real-world problems.
> 3. We agree that our tables of results are too aggregated, and that searching the big online archive of data is tedious. Therefore we add an appendix, comparing, in particular, YABBOB, MSYABBOB, ZPMSYABBOB, and also GYM problems for illustrating our conclusions on neurocontrol. We also add BBOB/COCO.

---

> ### Comment · Reviewer_gVy2 · 2024-04-15
> **Response to authors**
>
> Thank you for your clarification. However, my concern has not been addressed for the first point. As I mentioned in the original review, without using the proposed framework, one can test these properties by simply testing an algorithm with/without boundary, and by using different initial search distributions or initial search intervals. Therefore, it is not clear how this modification to the existing framework is particularly useful. Moreover, the importance of the distribution of the initial population is a widely known fact and so the claim itself, i.e., “scale matters” is not a novel finding.
>
> For the third point, thank you for reorganizing the results and additional results. However, as I pointed out in the original review, the analysis of the results is missing. It is nice to have a large number of promising results from the proposed approach, however, the analysis of the results is always necessary. The authors did not provide when it is promising and when it is not, and why these are the cases.

---

> ### Author Response · Authors · 2024-04-16
> **Automatizing tests vs manual tests and analysis**
>
> Regarding, << one can test these properties by simply testing an algorithm with/without boundary, and by using different initial search distributions or initial search intervals. >>:
> Thanks for your comment. Sure, we could run benchmarks several times with different parameters and aggregate results manually. However, the whole point of a benchmarking platform is to automatize the tests. It is certain that one can run BBOB or YABBOB several times and modify it manually each time and then average the results. However, people typically avoid such a work, and when they do it, they validate our principle, and it is much better for reproducibility to include the modification in the code. There is no platform with our simple scaling modifications, which has a clear big impact on the result.
> We added ablation studies for clarifying the impact of the modification, and our code is entirely automated so that unless people really hack the code their results with our benchmark will have to be robust to scaling issues if they want to get good results.
> We point out that there are several other conclusions, all included in NgIohTuned.
>
>
> << For the third point, thank you for reorganizing the results and additional results. However, as I pointed out in the original review, the analysis of the results is missing. It is nice to have a large number of promising results from the proposed approach, however, the analysis of the results is always necessary. The authors did not provide when it is promising and when it is not, and why these are the cases. >>
>
> In the recently revised version, each conclusion’s claim is now followed by the references to the figures and sections supporting them through the different analysis. For example, Fig. 2 shows excellent results for Carola / NgIoh, compared to a lot of baselines, whereas Carola is not that much better than other methods in the benchmarks without scaling issues (Fig 3 and ablation in appendix). We will be happy to add/clarify analyses for any specific point you might point out.
> We thank the reviewer once more for their insightful and positive review. We believe our paper is improved as a direct result of your comments. Should there be any more questions, please ask.

---

### Review · Reviewer_S4gr · 2024-03-14

**Summary Of Contributions:**

This paper analyzed existing benchmarks and observed the scaling of the optima significantly affects benchmarking results of optimization algorithms. Thus the authors proposed to adapt existing benchmarks with different scaling to make the benchmark more diverse. Based on the results of different optimization algorithms, the authors propose a new wizard NgIoh4 (a heuristic to switch between two existing methods) that performs the best in multi-scale benchmarks and across all the nevergrad benchmarks (not sure “all” means all the nevergrad benchmarks or not).

**Audience:**

Yes

**Broader Impact Concerns:**

No.

**Claims And Evidence:**

No

**Requested Changes:**

They are mostly covered in weakness. In summary:
- Have clear contribution statements with Nevergrad in context.
- Improve presentation quality as mentioned in the weakness.
- Include several examples that show clearly the impact of scaling to the existing benchmarks. This is needed to back up the first contribution. Currently it is indirect by showing that methods considering scaling perform better.

**Strengths And Weaknesses:**

On the strength, building diverse benchmarks is indeed an important aspect for the community. The empirical evaluations seem extensive.

On the weakness,

Many claims need to be supported. For example, the authors mentioned “many benchmarks have roughly the same norm of the optimum for all instances.” but I don’t find any supporting evidence from the paper. Also, the authors mentioned in the contribution “we observe that one can significantly modify the results of a benchmark by changing the distribution of the optima” but I can’t find support in the paper for it. The current support is rather indirect by showing methods that consider scaling performs better.

The presentation quality needs to be substantially improved.
* It is not clear to me what is the contribution on top of Nevergrad. Code is reported to be part of Nevergrad. Which parts of Nevergrad are specifically related to this work? The code at this link (benchmark/experiments.py) has contributions dated 2 years ago.
* The differences between all the variants of DE assumes familiarity of DE. The section 3.3.2 contains very dense content about many variants. It is impossible to follow unless the readers already know them. This is also true for other methods. The authors may choose important methods to explain and present in the paper rather than so many of them.
* In Section 4.2 and Section 5, why does the presentation of results become a list of how many times being the best for each method? I would expect something like Figure 2. Or at least like a bar plot. The current list of numbers weakens the presentation quality a lot.
* The language is not precise. Below are a few examples:
1. “adding BFGS after DE looks good”.
2. “when the ratio size/budget is not too low.”
3. “Scale is all you need”. This is a strong claim and on what ground the authors draw this claim?
4. “Good benchmarks exist…” What are the criteria for a good benchmark and why are the proposed benchmarks good benchmarks?
5. In Section 3.2.2, “Consequently, it reduces the generality of the conclusions.” What is the conclusion about and why does the norm of the minimum relate to that?

The results are currently hard to parse. It needs to be properly organized, selected or grouped so that the figures are more readable.

I am not sure if the inclusion of results from other external contributors have some potential intellectual property issues.

In the end, the novelty of the new wizard NgIoh4 is very limited.

(The section 6 may violate double blind requirements.)

---

> ### Author Response · Authors · 2024-03-26
> **Answer to review**
>
> We clarify our contributions:
> * In terms of code in Nevergrad, ZP-MS-BBOB and ZP-BBOB are our contributions, as well as quasi-opposite forms of DE and PSO, the modifications in Nevergrad for making the Gym environment operational again, and the Carola chaining, and all the NgIoh codes.  Quasi-opposite DE was already published, but the implementation in Nevergrad is ours, and our implementations of quasi-opposite forms of PSO in Nevergrad are new.
> * The authors of all other codes used in the paper (so-called “external” contributions) are all coauthors (the core authors proposed, they accepted, contributed, and validated the final paper).
>
> * Regarding the Nevergrad codebase, there are contributions which are 2 years old, but also much more recent contributions, and they appear on these files. In all the cases above, the contributions are ours, though the person who clicked on “squash and merge” on github or who copypasted some code in a PR might be a member of the Nevergrad team and not one of us.
>
> *DE variants: thanks for the suggestion, we added a pseudocode for QODE and clarified the list of variants. We also add a pseudo-code for SQOPSO.
>
> *Better support for the conclusions: we clarify the conclusion, and add references to appendices and to specific figures in the main part. Also, as suggested by another reviewer we added many results in the appendix, in a less aggregated form. The biggest change is adding NgIohTuned, for a better support of conclusions: whereas NgIoh4 was essentially focused on scaling issues in continuous domains, NgIohTuned adds the ability to take into account high-level information such as “real-world problem” or “neuro-control problem”: this incorporates all conclusions from the paper, and the performance gap NgIohTuned >> NGOpt is bigger than the performance gap NgIoh4 >> NGOpt. We include high-level figures about NgIohTuned in the main paper and appendix F with detailed results.
>
> *Presentation of results in 4.2/5:
> we switch to a graphical representation+add appendices with many detailed results.
>
>  *Thanks for pointing out unclear writing at several places in the text:
> -“ratio size/budget is not too low.” ⇐ we replace this poor statement by an explicit (numerical) statement.
> -“scale is all you need” ⇐ we replace this statement by “Scale matters”, and add more references.
> -In Section 3.2.2, “Consequently, it reduces the generality of the conclusions.” What is the conclusion about and why does the norm of the minimum relate to that?     ⇐ we modify the text and add references, for clarifying the point: conclusions drawn on benchmarks with a norm of the optimum almost always the same within a factor 2 are not valid for problems for which the norm of the optimum can vary by a factor 1000 (as in neuro-control, for example).
> -Impact of the scaling: we add an appendix exactly on this, showing how our chaining s are good on problems with unknown scale (based on Cobyla or quasi-opposite sampling)  solves this + many other figures in the appendix, in a less aggregated form for clarity.
>
> *Performance of NgIoh: our improvements, in NgIoh, are for cases that include continuous variables and cases in which the early stages are important, and for the rest NgIoh does not differ from NGOpt.leading to a small difference in global statistics. Our other conclusions (about NGOptRW for real-world problems and SQOPSO for neuro-control) have no impact on NgIoh. For solving this, we add a new NgIoh, namely NgIohTuned, which can use high level information such as 'this problem is real-world' or 'this variable is a neural weight. This makes the conclusions more visible, as it applies the successful rules 'use SQOPSO for neural weights' or 'use NGOPTRW for real-world problems'. Our new code is publicly available.
> We add an appendix showing that even on the old BBOB/COCO benchmark the performance is good.
>
> *Results of external contributors and intellectual property: thanks for caring about this. We care as well, which is why we included all authors of these codes as coauthors of the present paper, they validated it and nobody is forgotten.
>
> “Many benchmarks have roughly the same norm of the optimum for all instances” ⇒ we agree that we did not want to point out this or that benchmark having such problems, for not “attacking” papers which have been excellent contributions to the field. For making things clear, we explain in more detail in the revised version that the “nearly constant norm case” happens whenever some form of the central limit theorem applies: for example, the optimum is randomly drawn with independent laws for all coordinates, with roughly the same variance, but also if the optimum is randomly uniformly drawn in corners of a box.

---

> > ### Comment · Reviewer_S4gr · 2024-03-31
> > **Reponse to authors**
> >
> > As the authors write in the introduction, the contributions are two folds. The first contribution, also the more important one to me, is the analysis of the benchmarks. And two important findings include 1) the impact of scaling and 2) the special role of the center. These findings motivate designing of new variants of existing benchmarks. That's why I think it is critical that the authors providing strong evidence on the two observations. Currently the authors write many statements not supported by experimental results. Below I listed a few:
> >
> > "...Assuming that the optimum has all coordinates randomly independently drawn with center zero implies that the squared norm of the optimum is, nearly always, close to the sum of variances: this is the case in many artificial benchmarks..." I understand the rationale behind random coordinates in large dimension, but the authors claims it is the case in many benchmarks. Can the authors show what benchmarks are having this problem, and how many of them having such problem?
> >
> > "This is not observed in real-world benchmarks, hence the great real-world performance of the methods above (quasi-opposite sampling) tackling such issues” Can you show the scaling of many real world benchmarks (if the benchmark knows the optima)?
> >
> > "This special role of the center might imply that the neighborhood of zero provides too much information. Actually, many real-world problems have misleading values close to zero, in particular in control or neuro-control (e.g., for neuro-control the control is just zero if all weights in a layer are zero).” What do authors mean by "too much information" and "misleading values"? Can you show with experimental results of "many real-world problems"?
> >
> > The authors also write in the reply that "..we agree that we did not want to point out this or that benchmark having such problems, for not “attacking” papers which have been excellent contributions to the field…". I don't think showing problems of existing benchmarks or methods is attacking other papers. Claiming but not providing convincing evidence is not fair.
> >
> > On the algorithmic contribution, the main contributions are NgIoh and NgIohTuned (I may miss some). Can the authors make it clear in the introduction section so the readers know what they should look for? How do the proposed algorithm performs compared with other baselines in short? There are so many baselines in the results, what are the most relevant one to look at? I find it difficult to navigate through the results. Also, the results seem not conclusive, this is also pointed out by other reviewers. What's more, how does the Section 6 relates to the contribution of this paper? I don't see the connection to the problems of scaling, center, and NgIoh.
> >
> > The authors fixed some places I mentioned but not beyond. Overall, I think the paper needs a major revision.

---

> > > ### Author Response · Authors · 2024-04-04
> > > **Benchmarks**
> > >
> > > Thanks for your review. We did a major revision as requested.
> > >
> > > Short answer: we move a lot of things for clarifying, we now have all the key results in Fig. 1, 2, 3, 4, and Fig 3 (right) shows the strength of our new wizard NgIohTuned which aggregates all the outcomes of the paper (whereas in previous versions we were only using the scaling issues for creating NgIoh4 as an improvement compared to NGOpt). We get much better results (Fig. 3 right shows how much NgIohTuned dominates). This is now reflected in the title.
> > >
> > > Detailed answer:
> > > Benchmarks with coordinates of optimum having such properties: for example, this is the case for COCO/BBOB, which is the most widely used: the optimum always has a distance to the center of the same order as the distance to the bounds. We are not aware of a single benchmark in which we can have, as in our proposed benchmarks, an order 1000 between the norms of optima from instance to instance of a same test case (thanks to a random factor scaling from 0.01 to 10), even within the 90% confidence interval.. A more controversial point is that we observe papers in which the different methods compared in the code have completely different initial norms: this is certainly a risky practice, and baselines are useless if they work at scale 10.0 whereas the right scale is 0.01 and only the advocated method is run at that scale.
> > >
> > > Importance of the scaling in real-world benchmarks: for example, in neural networks, the very well known Xavier initialization explains that the scale varies depending on the fan in and fan out, and in many cases there is a big gap at the optimum between the absolute values of different weights. More recent papers about initialization scales in neural networks (e.g. https://arxiv.org/abs/1704.08863 ) explain that this is a first approximation and varies a lot. So, users can not know how far from the origin the optimum is.
> > > Also, see real-world problems in our paper: what are reasonable bounds for the parameters in Fishing, Aquacrop or Photonics ? This is quite arbitrary and in many cases, we will be far from the bound, which is a kind of exceptional case, and the bounds are not at all the same for the different parameters.
> > > Also, we would say that the burden of proof is on the other side: the optimization method should not assume that necessarily the optimum is at a distance of the center of the same order as the distance to the bounds. Note that when there is no bound (unconstrained optimization, as in many neurocontrol problems unless a quite arbitrary bound is chosen by the user), the same issue arises: we are just not aware of a single benchmark which has this principle of unknown scale of the optimum, and in which the norm of the optimum varies (even in terms of 95% confidence intervals) by a factor 1000. We provide additional references in Section 3.2.2. Also, our results (Fig. 4 left, for the real world, and Fig. 2, for our multi-scale benchmarks) show that in both cases, we get good results for quasi-opposite methods and for a warmup by Cobyla (see Carola3).
> > >
> > > Regarding the special role of zero in neurocontrol cases: it is known that at zero, neural networks just always return zero and the derivative with respect to each parameter is null. There are symmetries as soon as we have at least two layers, see e.g. discussions in Bishop’s books. Therefore, there is little information in terms of direction towards the optimum. For other types of policies, with coefficients corresponding to the intensity of the answer (as in parametric rules), zero also leads to a null control.

---

> > > > ### Author Response · Authors · 2024-04-04
> > > > **Contributions, and Section 6 (problems by external contributors) now moved to appendix**
> > > >
> > > > Algorithmic contributions: we modify the following in the introduction:
> > > > <<
> > > > In \Cref{sec:algos}, we focus on algorithms which perform well in a context of unknown scale, both in discrete and continuous domains, and in real-world scenarios.
> > > > These contributions are integrated into a state-of-the-art wizard for black-box optimization which improves the state of the art on average on many benchmarks (\Cref{XX} and later): NgIoh4, which incorporates our improvements regarding scale issues, and NgIohTuned, which incorporates all the modifications that we propose, including switching to quasi-opposite PSO in neuro-control and to NGOptRW for other real-world problems.
> > > > >>
> > > > We also clarify the last sentence of the abstract:
> > > > <<All methods are included in a public optimization wizard, namely NgIoh4 (without taking into account the type of variables) and NgIohTuned (taking into account all conclusions of the paper, including taking into account the real-world nature of a problem and/or that it is neurocontrol).>> and we update the title accordingly: we do this because we feel that at the end NgIohTuned is the clear synthesis in a single code of all the paper.
> > > >
> > > > Regarding Section 6, the whole point is to confirm our conclusions regarding real-world problems on completely distinct benchmarks (i.e. benchmarks that we have never seen when designing NgIoh). We confirm in Section 6 the conclusions regarding SQOPSO (in particular on Gym), regarding DE (all continuous problems), and the performance of Lengler’s method in the discrete case: Section 6 confirms the rest of the paper.
> > > > We move it to the appendix, so that it does not break the flow of the article. Also, as mentioned above, they are consistent with the problems of scaling: bounds in continuous cases (we did not modify them) are chosen quite heuristically, as in many real-world problems.
> > > >
> > > > The beginning of Section 6 is as follows:
> > > > <<Finally, we include a few use cases by Nevergrad users. The benchmarks and setups have been developed independently of the benchmarking platform included in Nevergrad. The plotting tools, functions, and criteria, are frequently different from the rest of the paper. They, on purpose, quantify the robustness of the conclusions drawn on our update of the Nevergrad benchmark, specifically for the real-world cases. Overall,
> > > > results in Sections 6.1, 6.2, and 6.4 confirm the conclusion, in Nevergrad benchmarks, that DE performs well on many real-world problems; the discrete problem in Section 6.3 confirms the good performance of Lengler though FastGA(Doerr et al., 2017) is also good; Section 6.5 confirms the performance of SQOPSO when the scale of the optimum is unknown, in particular in the neuro-control case. >>
> > > > (we removed a minor conclusion, for clarity)

---

> > > > > ### Author Response · Authors · 2024-04-04
> > > > > **Major revision needed: adding NgIohTuned and rewriting abstract/intro/conclu for emphasizing it as an aggregation of all conclusions of the paper**
> > > > >
> > > > > Regarding the changes in the revision, the conclusion and many parts have been updated. In particular the conclusion contains references to sections and figures justifying them. Also, NgIohTuned is entirely new in the revision: we created it so that it includes all the conclusions of our paper. As stated in the conclusion: <<NgIohTuned, using high-level information on the type of problem and the types of variables, outperforms other wizards and aggregates in a single code all the conclusions in the present section.>> Following your review, we clarify this in the introduction and abstract. This also answers comments about the importance of our results: thanks to this more powerful wizard (including all our conclusions, and not only the part about scaling), we get a bigger gap than with the previous wizard.

---

> ### Author Response · Authors · 2024-04-11
>
> If we understand correctly this is the last day for us to address concerns: therefore we submit a revision.
> 1) We believe NgIohTuned addresses "Have clear contribution statements with Nevergrad in context.", as NgIohTuned is a code which takes into account all our conclusions, and not only the scaling part, so that there is a single code clearly. We believe  that
>    - wizards which do algorithm selection based on conclusions of papers
>    - and big benchmarks (we have significantly increased the Nevergrad benchmarking platform, arguably the current biggest in black-
>       box optimization).
> are stable and reproducible tools for improving the state of the art.
> 2) Our previous revision was vast, hopefully addressing "Improve presentation quality as mentioned in the weakness".
> 3) Regarding the comment "Include several examples that show clearly the impact of scaling to the existing benchmarks. This is needed to back up the first contribution. Currently it is indirect by showing that methods considering scaling perform better.", we add a section in appendix "Ablation regarding ZP and MS: the importance of scaling in continuous domains", referenced in the main text.

---

### Decision · Action_Editor_p1SF · 2024-04-24

**Recommendation:** Reject

**Comment:**

I read this paper in detail and totally agree with all reviewers that the proposed benchmarks are valuable, and I also appreciate the extensive empirical evaluations and the public benchmark platform and optimization wizard. In addition to Meunier et al. and Dagstuhl participants 2023, there are also some recent works that discuss the importance for reliable benchmarks and fair comparison for black-box optimization [1,2,3]. A relative center-bias issue has been investigated in [4,5]. Therefore, if all claims are well supported, this work could be very helpful and potentially impactful for the community.

However, according to the reviewers' official recommendation, some major claims are still not well supported by evidence, and the current paper requires a major revision and substantial improvement. Since TMLR does not support a direct major revision option, I suggest rejection for the current submission but encourage the authors to **consider submitting a major revision at a later time**. If the authors choose to prepare a major revision, I hope the quoted remaining concerns from the reviewers will be carefully addressed.

[1] Jerry Swan, Steven Adriaensen, Alexander EI Brownlee, Kevin Hammond, Colin G. Johnson, Ahmed Kheiri, Faustyna Krawiec, J. J. Merelo,
Leandro L. Minku, Ender Ozcan, Gisele L. Pappa, Pablo Garcia-Sanchez, Kenneth Sorensen, Stefan Voss, Markus Wagner, David R. White. Metaheuristics "in the large". European Journal of Operational Research 297, no. 2 (2022): 393-406.

[2] Nikolaus Hansen, Anne Auger, Dimo Brockhoff, and Tea Tusar. Anytime performance assessment in blackbox optimization benchmarking. IEEE Transactions on Evolutionary Computation 26, no. 6 (2022): 1293-1305.

[3] Thomas H.W. Back, Anna V. Kononova, Bas van Stein, Hao Wang, Kirill A. Antonov, Roman T. Kalkreuth, Jacob de Nobel, Diederick Vermetten, Roy de Winter, and Furong Ye. Evolutionary algorithms for parameter optimization—thirty years later. Evolutionary Computation 31, no. 2 (2023): 81-122.

[4] Jakub Kudela. A critical problem in benchmarking and analysis of evolutionary computation methods. Nature Machine Intelligence 4, no. 12 (2022): 1238-1245.

[5] Jakub Kudela. The evolutionary computation methods no one should use. arXiv preprint arXiv:2301.01984 (2023).

**Audience:**

All reviewers believe some individuals in TMLR's audience could be interested in the findings of this paper.

**Claims And Evidence:**

This work investigates black-box optimization, especially the gap between current artificial benchmarks and real-world application problems. The key contributions are mainly two-fold: 1) The authors identify and analyze an important scale issue that significantly affects different algorithms' performance on the current benchmark problems. To address this issue, they propose a set of benchmarks with different scales, which also carefully takes reproducibility into consideration. 2) The authors further propose a new black-box optimization wizard called NgIohTuned that achieves robust and promising performance on average in many benchmarks.

The reviewers appreciate the new benchmark problems proposed in this work, as well as the extensive empirical evaluations, the promising performance achieved by the proposed NgIohTuned, and the publicly available benchmark platform. They also raise valuable concerns and constructive criticism about the not-well-supported claims in the submitted paper. The authors have provided a revision with a detailed response to address the raised concerns. However, after rebuttal, two reviewers still believe their concerns are not well addressed and some claims in this work are not well supported by accurate, convincing and clear evidence.

 **Reviewer S4gr's Remaining Concerns:**

In the official recommendations, Reviewer S4gr would like to thank the authors for their reply and does see there are some improvements regarding the comments. Reviewer S4gr thinks "Figure 5 in Appendix A is a good example to show considering different scaling in the benchmarks makes the ranking of the method change. And I think it should be highlighted in the main text before introducing the multi-scaling benchmarks. The part about zero penalizing is not so clear from Figure 5 though." However, this reviewer also has some major concerns about the current paper. Since the comment in the official recommendation is not visible to the authors, I quote the concerns by Reviewer S4gr here:

>"I think the paper needs some structural change: I would suggest separating the algorithm contribution from the observations. Currently 3.3.1-3.3.3 are observations from the benchmarks and 3.3.4 and 3.4 are the proposed algorithms based on the observations. I would suggest moving 3.3.4 to Section 3.4 to make it clear. Or even better move the 3.3.4 and 3.4 into a whole new section just to make the logical connection clear between deriving the observations (Section 3) and proposing improvement (a new section). Also, the current Section 4 title should be “Results” and Section 5 should be part of it."

>"For presenting the results, when reading Figures 2,3 and 4, because there are so many variants of the proposed algorithms, what are the main baselines we should look at? The authors should guide the readers. Also, what are NgIoh4 and NgIoh6? What is the difference compared with NgIoh4? Also, if NgIohTuned includes NgIoh4, should the author only present results for NgIohTuned? It will make the reading of the results easier."

>"In the end, NgIoh4 is a heuristic (based on benchmarking results) switching between NGOpt and Carola2, which again heuristics for switching Cobyla, CMA and SQP. The other algorithm contribution, NgIohTuned, again switches among NgIoh4, NGOptRW and SQOPSO, based on the property of the problems (if it is a real-world, uses NGOptRW; if it is a neural control problem, uses SQOPSO). One may argue where is the “learning” perspective for this algorithm and the generalization ability of the proposed methods."

**Reviewer gVy2's Remaining Concerns:**

In the official recommendation, Reviewer gVy2 appreciates the large number of experimental results, but still believe:

> "However, the analysis of experimental results is missing. Therefore, the usefulness of the approach is questionable. This point was not addressed in the revision."

>"It is also not clear how the proposed modification of the existing benchmarking is useful. As pointed out in my review, the importance of the initial distribution of the population is a widely known fact and is easy to verify without the proposed framework. This point has also not been addressed in the revision."

**Resubmission Of Major Revision:**

The authors may consider submitting a major revision at a later time.